# Independent manipulation of histone H3 modifications in individual nucleosomes reveals the contributions of sister histones to transcription

Zhen Zhou[1†], Yu-Ting Liu[1†], Li Ma[2], Ting Gong[1], Ya-Nan Hu[2], Hong-Tao Li[1], Chen Cai[1,3], Ling-Li Zhang[1], Gang Wei[2], Jin-Qiu Zhou[1,3]*

[1]State Key Laboratory of Molecular Biology, CAS Center for Excellence in Molecular Cell Science, Innovation Center for Cell Signaling Network, Shanghai Institute of Biochemistry and Cell Biology, University of Chinese Academy of Sciences, Chinese Academy of Sciences, Shanghai, China; [2]Key Laboratory of Computational Biology, CAS-MPG Partner Institute for Computational Biology, Shanghai Institutes for Biological Sciences, Chinese Academy of Sciences, Shanghai, China; [3]School of Life Science and Technology, Shanghai Tech University, Shanghai, China

**\*For correspondence:**
jqzhou@sibcb.ac.cn

[†]These authors contributed equally to this work

**Competing interests:** The authors declare that no competing interests exist.

**Abstract** Histone tail modifications can greatly influence chromatin-associated processes. Asymmetrically modified nucleosomes exist in multiple cell types, but whether modifications on both sister histones contribute equally to chromatin dynamics remains elusive. Here, we devised a bivalent nucleosome system that allowed for the constitutive assembly of asymmetrically modified sister histone H3s in nucleosomes in *Saccharomyces cerevisiae*. The sister H3K36 methylations independently affected cryptic transcription in gene coding regions, whereas sister H3K79 methylation had cooperative effects on gene silencing near telomeres. H3K4 methylation on sister histones played an independent role in suppressing the recruitment of Gal4 activator to the *GAL1* promoter and in inhibiting *GAL1* transcription. Under starvation stress, sister H3K4 methylations acted cooperatively, independently or redundantly to regulate transcription. Thus, we provide a unique tool for comparing symmetrical and asymmetrical modifications of sister histone H3s in vivo.
DOI: https://doi.org/10.7554/eLife.30178.001

## Introduction

In eukaryotes, chromatin carries both genetic and epigenetic information that controls multiple cellular processes, such as DNA replication, transcription and genome organization (*Berger, 2007*; *Lawrence et al., 2016*; *Papamichos-Chronakis and Peterson, 2013*). The basic unit of chromatin is the nucleosome, which comprises ~147 bp of DNA and a histone octamer formed by two copies of histone H2A-H2B and H3-H4 heterodimers (*Bentley et al., 1984*; *Kornberg and Thomas, 1974*; *Luger et al., 1997*; *Oudet et al., 1975*). The packaging of DNA into nucleosomes affects sequence accessibility, and nucleosomes therefore regulate the activity of DNA-binding proteins (*Lee et al., 1993*; *Wasylyk and Chambon, 1979*). Histones also appear to protect DNA from breaking and maintain the fidelity of both replication and transcription (*Carrozza et al., 2005*; *Govind et al., 2007*; *Joshi and Struhl, 2005*; *Keogh et al., 2005*; *Pinskaya et al., 2009*). The regulation of nucleic acid metabolism by nucleosomes is mediated through multiple post-translational modifications (PTMs), such as methylation, acetylation, phosphorylation, and sumoylation (*Lawrence et al., 2016*).

Histone lysine methylation, especially on histone H3, regulates chromatin structure and transcription (*Ng et al., 2002*; *Vermeulen and Timmers, 2010*; *Wagner and Carpenter, 2012*). In budding

**eLife digest** Inside each human cell, about two meters of DNA is wrapped around millions of proteins called histones, forming structures known as nucleosomes. Each nucleosome contains 147 letters of DNA code and two copies of four different histones – H2A, H2B, H3 and H4 – meaning eight proteins in total.

The two copies of each histone protein found in a nucleosome are referred to as "sister" histones and are identical. Histone proteins have long tails that the cell can edit by adding chemical groups at specific positions. This changes the way the cell copies, uses and repairs its DNA. Previous studies show that identical sister histones can end up with different modifications. But, it was not clear what effect this had.

To adress this issue, there are two questions to answer. What do asymmetric sister histones do in living cells? And, does a modification to one histone affect its sister? Gene editing could help scientists to understand the effect of asymmetrical tail modification by forcing cells to make non-identical sister histones. However, this is challenging because most animals studied in the laboratory have many copies of the genes for histones. Fruit flies, for example, have 23 copies of their histone genes. The single-celled yeast *Saccharomyces cerevisiae* has only two copies of its histone genes. Yet, even if one of these genes was replaced with a mutant gene and the other left unedited or "wild-type", there would be nothing to stop the cell from forming nucleosomes in which both sister histones were still identical – that is to say, mutant with mutant or wild-type with wild-type.

Now, Zhou, Liu et al. report a new method that allowed them to edit the tail sequence of one H3 histone but not its sister. First, they searched for, and found, a pair of mutant H3 genes, which encode two extremely similar but different H3 proteins that could bind to each other but not to themselves. As a result, yeast cells with the genes for these proteins could only form nucleosomes in which the sister H3 histones were non-identical. Next, Zhou et al. made a small change to the tail of one of the H3 sisters which meant it could not be modified. The resulting nucleosomes contain one H3 histone with a wild-type tail and one with a mutant tail. The cell could only modify one of them, mimicking natural asymmetrical modifications.

The new technique revealed that modification of one sister does not affect the the other. It also revealed that modifications to sister histones can work both alone and together. In some cases, the cell needs only edit one tail to affect the use of a gene. Other times, it must edit both tails for greatest effect.

This new tool is the first step in understanding the contribution of the tails of sister histones in living cells. In future, it should help to uncover the effect of different combinations of modifications. This could shed light on how cells control the use of different genes.

DOI: https://doi.org/10.7554/eLife.30178.002

yeast, the best-studied methylations on histone H3 are methylation of lysine at amino acid positions 4, 36, and 79 (H3K4, H3K36 and H3K79, respectively). H3K4 di- and tri-methylation (H3K4me2/3) is catalyzed by the Set1 complex (also called the COMPASS complex) and is associated with steady-state gene transcription; thus, H3K4me2/3 is considered to be an 'activating' mark in mammals. Conversely, in budding yeast, most of the evidence indicates that H3K4 methylation is a repressive mark (*Shilatifard, 2006*; *Weiner et al., 2012*). H3K36 tri-methylation (H3K36me3) by Set2 directs the deacetylation of histones, predominantly at the 3' portion of gene open reading frames (ORFs), to suppress spurious intragenic transcription initiation (*Carrozza et al., 2005*). Methylation of H3K79 (H3K79me) affects telomeric heterochromatin structure because mutations at H3K79 as well as inactivation of its methyltransferase, Dot1, lead to loss of telomere silencing (*Jones et al., 2008*; *Ng et al., 2002*). The functions of each modification are largely dissected by using histone mutations in combination with the inactivation of corresponding methyltransferases, under which circumstances the modifications on both sister histones are simultaneously removed, making it difficult to study the crosstalk between modifications on sister histones.

Although two copies of each histone in a nucleosome possess identical protein sequences, histone-modification enzymes do not always modify sister histones simultaneously (*van Rossum et al., 2012*; *Voigt et al., 2013*). For example, symmetrical modification of histone lysines within a

nucleosome is not globally required in HeLa cells (*Chen et al., 2011*). In addition, in different cell types, a significant number of nucleosomes contain asymmetrically modified sister histones (*Fisher and Fisher, 2011*; *Mikkelsen et al., 2007*; *Voigt et al., 2012*). Furthermore, asymmetrically modified nucleosomes are present in embryonic stem cells but are symmetrically modified upon differentiation (*Voigt et al., 2012*). Each of these studies suggests that sister histones within a single nucleosome may function independently in gene regulation. A synthetic system for the generation of asymmetrically modified nucleosomes has been used to study histone PTM crosstalk in vitro (*Lechner et al., 2016*), but the lack of a genetic model system for studying asymmetric histone modifications in vivo has prevented exploration of the biological significance of this previously documented phenomenon.

To investigate the individual contributions of sister histones and their modifications to chromatin structure and function, we employed a protein engineering strategy to mutate both copies of histone H3 in their interaction interface. After screening for mutants that were able to form histone H3 heterodimers but not H3 homodimers, we successfully set up a bivalent nucleosome system in the budding yeast *Saccharomyces cerevisiae*. By using this unique system, we validated the establishment of asymmetrically methylated H3K4, H3K36 or K3K79 in chromatin in yeast in vivo. Furthermore, we examined the functions of asymmetrically modified sister histones in the regulation of chromatin structure and gene transcription. Our results revealed that modifications such as H3K4me, H3K36me or K3K79me on sister histone H3s could be independent of each other. In addition, the same modifications on both sister H3 histones can affect transcription in a cooperative, independent or redundant manner. Our study provides the first picture of the individual contributions of sister histones to chromatin dynamics in vivo.

## Results

### A bivalent nucleosome system to study sister histone H3s in yeast

In *S. cerevisiae*, each canonical histone is encoded by two genes. H3 is encoded by *HHT1* and *HHT2*, and H4 is encoded by *HHF1* and *HHF2*. The histone genes are organized into a pair of divergently transcribed loci with *HHT1-HHF1* and *HHT2-HHF2* linked together. Owing to redundancy, deletion of either locus does not cause lethality (*Dollard et al., 1994*). As asymmetrical modifications have previously been reported on histone H3 in vivo (*Voigt et al., 2012*), we began by examining H3. Previous structural work revealed that two molecules of histone H3 interact through their carboxy-terminal four-helix bundle to form a homodimer (*Luger et al., 1997*; *Ramachandran et al., 2011*; *White et al., 2001*) (*Figure 1A*). We performed site-directed mutagenesis on the Ala110, Ala114 and Leu130 residues of the *HHT1* gene. These residues were chosen because they were spatially close and within the bundle region that interacts to form the H3 homodimer (*Luger et al., 1997*; *Ramachandran et al., 2011*; *White et al., 2001*). These neutral amino acids were mutated to acidic or basic residues to make them electronegative or electropositive under physiological conditions. We reasoned that if we created an H3 allele with an electronegative (or electropositive) interface, it would not form homodimers, but it would interact with a different H3 allele with an electropositive (or electronegative) interface, thereby creating a heterodimer (*Figure 1A*).

Yeast cells lacking chromosomal *HHT1* and *HHT2* genes but containing the *HHT1* gene on a counter-selectable *URA3* plasmid were transformed with plasmids carrying the mutated histone H3 genes. We then screened for histone H3 mutants that did not support cell viability when loss of the wild-type (WT) *HHT1* gene was counter-selected using 5-fluoroorotic acid (5-FOA) (*Figure 1B*). Only the H3 mutant bearing the A110E mutation survived (*Figure 1C*), suggesting that the other 14 histone H3 mutants could not form a homodimer. Next, yeast cells were co-transformed with the pairwise plasmids carrying these 14 mutated histone H3 genes (*Figure 1D*). Notably, only the H3A110D and H3L130H pairing was able to support cell viability on a 5-FOA plate (*Figure 1E*; *Figure 1—figure supplement 1*), allowing us to infer that the H3A110D and H3L130H mutants form a heterodimer that could be assembled into functional nucleosomes in vivo. Considering that histidine's positive charge is weakened under physiological pH conditions and may increase the risk for H3L130H self-interaction, we used the relatively weaker *ADE3* promoter to reduce the expression of H3L130H (*Agez et al., 2007*; *Antczak et al., 2006*). This strain will be hereafter referred to as the H3$^D$/H3$^H$ strain.

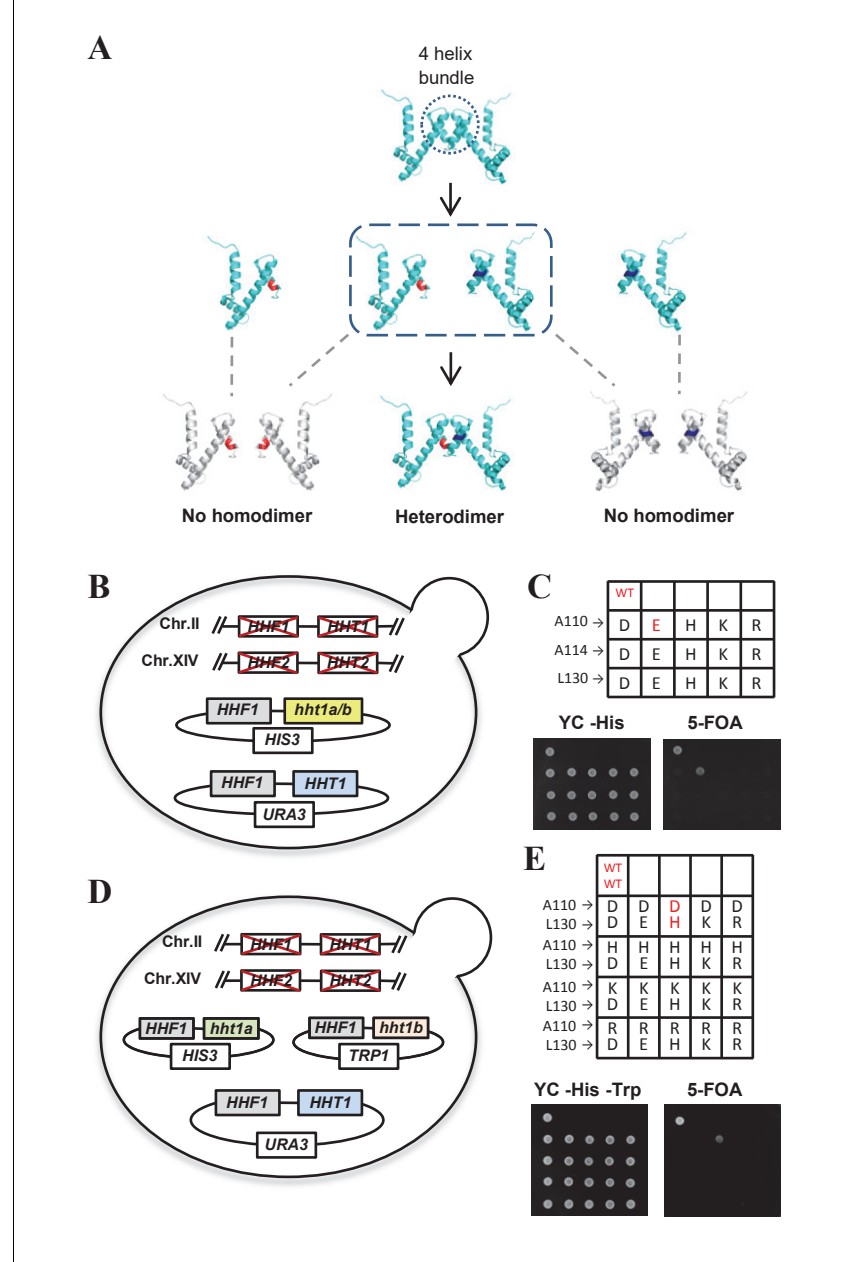

**Figure 1.** The complementary mutants of histone H3A110D and H3L130H are assembled into nucleosomes in vivo. (**A**) Schematic illustration of the production of histone H3 mutants that are able to complement and form heterodimers. (**B and D**) Schematic strategy for screening histone H3 mutants. Genomic *HHF1*, *HHF2* (encoding histone H4), *HHT1* and *HHT2* (encoding histone H3) genes were knocked out, and yeast cells (LHT001 background) were transformed with plasmids expressing wild-type (WT) or mutated histone H3 along with WT H4. (**C**) Histone H3 mutants cannot survive without WT histone H3. Dotting assay was performed to examine the cell viability of WT cells (LHT001) and 15 histone H3 mutants carrying pRS313-*hht1a/b* (*HIS3* marker). (**E**) H3A110D and H3L130H mutants can form a heterodimer. Mutants carrying both pRS313-*hht1a* (*HIS3* marker) and pRS314-*hht1b* (*TRP1* marker) were dotted on a 5-fluoroorotic acid (5-FOA) plate. WT and mutants that survived on 5-FOA plate are highlighted in red.

DOI: https://doi.org/10.7554/eLife.30178.003

The following figure supplement is available for figure 1:

**Figure supplement 1.** The pairwise histone H3 mutants other than H3A110D and H3L130H cannot grow when the WT histone H3 gene is counter-selected using 5-FOA.

DOI: https://doi.org/10.7554/eLife.30178.004

## Characterization of the H3$^D$/H3$^H$ strain

To confirm that mutant histones H3A110D and H3L130H equally assembled into nucleosomes, we epitope-tagged one copy of H3 in H3$^D$/H3$^H$ cells with Myc. After preparing mono-nucleosomes (*Figure 2—figure supplement 1*), we performed immunoprecipitations with an anti-Myc antibody and examined both Myc-tagged and untagged histone H3. In the control, the chromatin from both the myc-H3 strain and the untagged H3 strain was mixed, and the immunoprecipitation of mononucleosomes using the anti-Myc antibody did not pull down untagged H3 (*Figure 2A*, second lane). As the anti-H3 N-terminal antibody could not recognize Myc-tagged histone H3 (*Figure 2A*), we normalized immunoprecipitated myc-H3L130H and myc-H3A110D to the same level. The amounts of the co-immunoprecipitated complementary H3A110D and H3L130H histones were identical (*Figure 2A*), reflecting an equal incorporation of H3A110D and H3L130H into mononucleosomes in H3$^D$/H3$^H$ cells. Next, we examined the ratio of H3A110D to H3L130H and the nucleosome positioning at the *GAL1-10* gene locus in the H3$^D$/H3$^H$ cells. *GAL1-10* intergenic chromatin consists of a non-nucleosomal, UAS-containing hypersensitive region (*Lohr, 1984*; *Lohr and Hopper, 1985*) surrounded by positioned nucleosomes (*Lohr and Lopez, 1995*; *Lohr et al., 1987*). A chromatin immunoprecipitation (ChIP) assay showed almost the same enrichment of H3A110D and H3L130H at the *GAL1* gene promoter (*Figure 2B*), supporting our conclusion that mutant histones H3A110D and H3L130H were assembled into nucleosomes at a ratio of 1:1 in vivo. MNase digestion of the *GAL1-10* promoter revealed that the nucleosome array on the *GAL10* side of the UAS region displayed a similar digestion pattern in H3$^D$/H3$^H$ and WT cells, but the nucleosome array on the *GAL1* side showed a more evenly digested pattern in WT cells than in H3$^D$/H3$^H$ cells (*Figure 2C*), suggesting altered nucleosome stability in the *GAL1* region in H3$^D$/H3$^H$ cells.

We next determined the functional viability of the H3$^D$/H3$^H$ mutant using the histone shuffle strain (LHT001) as a WT control. H3$^D$/H3$^H$ mutant and WT cells exhibited identical growth rates in yeast extract peptone dextrose (YPD) medium at 23°C, 30°C and 37°C. In addition, when H3$^D$/H3$^H$ cells were challenged by rapamycin (data not shown) or DNA-damage reagents, such as phleomycin or methyl methanesulfonate (MMS), they showed nearly the same sensitivity as WT cells (*Figure 2D*). Interestingly, compared with WT cells, H3$^D$/H3$^H$ cells showed a reduced growth rate when cultured in raffinose or glycerol medium (*Figure 2E*). We then checked the levels of multiple histone PTMs in WT and H3$^D$/H3$^H$ strains by western blot and found no significant differences (*Figure 2F*). Further, we performed a genome-wide RNA-Seq assay to examine the gene expression profiles in WT and H3$^D$/H3$^H$ strains. Statistical analysis confirmed the reproducibility of the RNA-Seq results in each strain (*Figure 2—figure supplement 2*). The global gene expression profile of the H3$^D$/H3$^H$ strain was found to be very similar to that of the WT strain (*Figure 2G*), but we did see some genes with expression levels that varied between the H3$^D$/H3$^H$ and WT strains. Through Gene Ontology analysis (see the Materials and methods for details), we found that most of the outliers were downregulated by histone H3 mutations. Interestingly, the genes encoding cytochrome-c reductase activity and ATPase activity were among the outliers (*Tzagoloff et al., 1975*) (*Figure 2—source data 2*). This finding might provide an explanation for the reduced growth of the H3$^D$/H3$^H$ strain when glycerol was used as the carbon source (*Figure 2E*).

Taken together, the observations presented in *Figure 2* indicated that the H3$^D$/H3$^H$ strain behaved similar to the WT strain under most, but not all of the tested circumstances; thus, this strain provides a unique and valid system for analyzing asymmetrically modified sister histones.

## N-terminal deletion of one sister histone H3 tail does not affect the other tail

To address whether there is crosstalk between the amino-terminal tails of sister histone H3s in one nucleosome, we constructed strains that lacked the N-terminal 4–15 amino acids on one or both sister H3 histones (*Mann and Grunstein, 1992*). The H3$^D$Δ4–15/H3$^H$ and H3$^D$/H3$^H$Δ4–15 strains contained one copy of N-terminal-deleted H3, resulting in asymmetrically deleted histone H3 (*Figure 3A*). The H3$^D$Δ4–15/H3$^H$Δ4–15 strain containing two copies of N-terminal-deleted H3 was also constructed and used as a negative control. The nucleosomes of the H3$^D$/H3$^H$ (treated as WT hereafter) and mutant strains were precipitated, and western blots were performed to examine the levels of histone H3 N-terminal and K4 tri-methylation. Both histone H3 N-terminal and H3K4me3 signals in H3$^D$Δ4–15/H3$^H$ and H3$^D$/H3$^H$Δ4–15 cells were reduced to approximately half of those

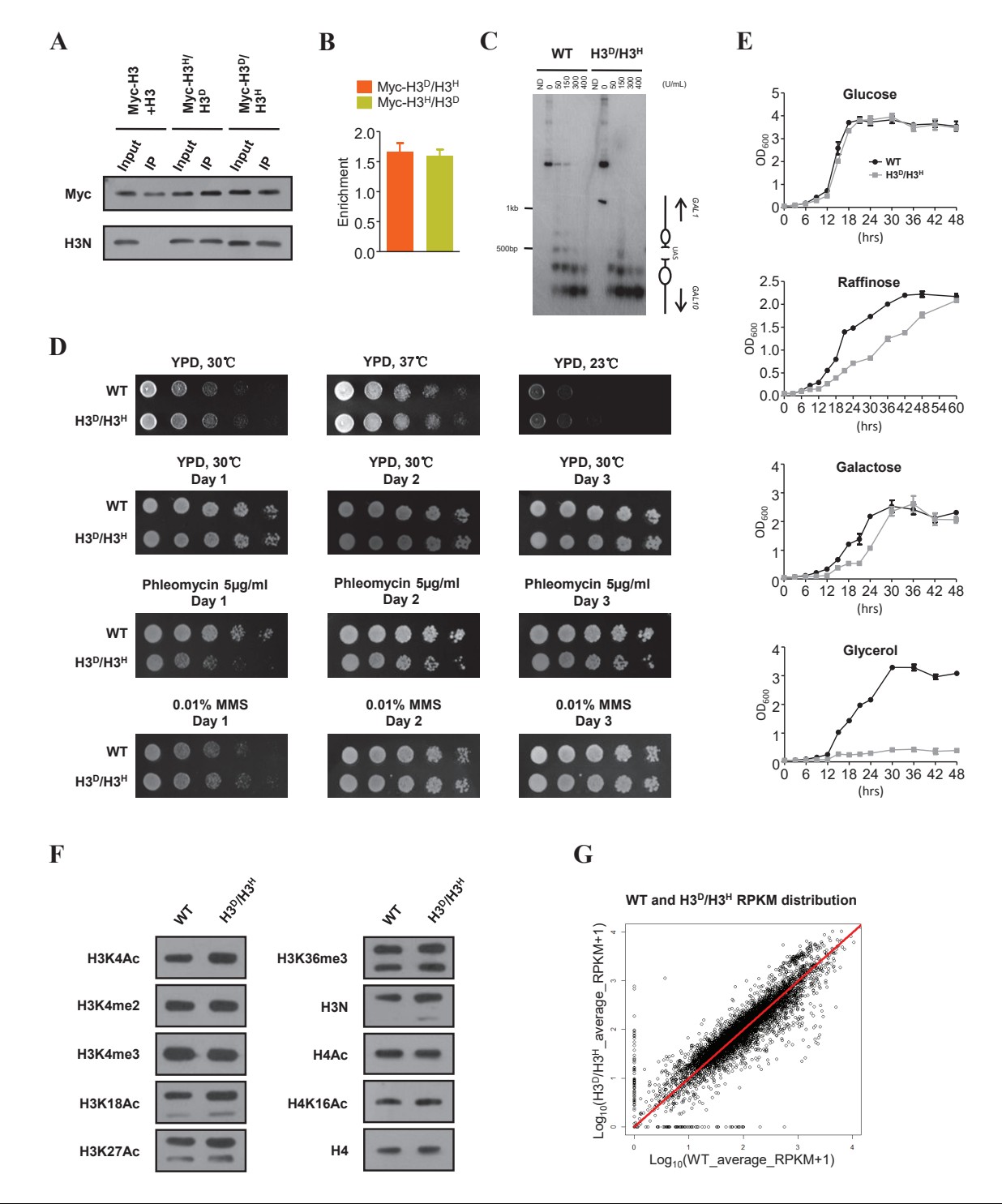

**Figure 2.** Characterization of the H3$^D$/H3$^H$ strain. (**A**) Mononucleosomes prepared from both the cells expressing Myc-tagged H3 (ZL38) and the cells expressing untagged H3 (LHT001) were mixed (Myc-H3 +H3) and immunoprecipitated with the anti-Myc antibody. The same immunoprecipitation (IP) assay was performed with mononucleosomes from strains bearing Myc-H3L130H/H3A110D (Myc-H3$^H$/H3$^D$) and Myc-H3A110D/H3L130H (Myc-H3$^D$/H3$^H$). The precipitated mononucleosomes were denatured and examined by western blotting with both anti-Myc (Myc) and anti-histone H3 N-terminal antibodies (H3N). (**B**) Chromatin IP (ChIP) analysis was performed in Myc-H3A110D/H3L130H (Myc-H3$^D$/H3$^H$) and Myc-H3L130H/H3A110D (Myc-H3$^H$/H3$^D$) cells using anti-Myc antibody. The precipitated DNA was analyzed by qRT-PCR with primers specific for the *GAL1-10* gene promoter and normalized to the *ACT1* gene. Error bars indicate s.e.m. for three independent experiments. (**C**) MNase digestion of nuclei from WT (LHT001) and H3$^D$/H3$^H$ strains. Nuclei were digested with increasing concentrations of MNase for 4 min. MNase cleavage sites were mapped from the EcoRI site within

*Figure 2 continued on next page*

*Figure 2 continued*

*GAL10* by indirect end labeling analysis on a 1.6% agarose gel. Marker fragments are from PCR products of 1 kb and 500 bp in length. The UAS region and nucleosome positions are schematically shown on the right. ND, naked DNA. (D) Dotting assays were performed in H3D/H3H mutant and WT (LHT001) cells. Plates were photographed after incubation at 37°C, 30°C and 23°C on yeast extract peptone dextrose (YPD) medium or after incubation at 30°C on YPD, YPD containing phleomycin and YPD containing MMS on Days 1, 2 and 3. (E) Growth curve assays were performed in H3D/H3H mutant and WT cells for the indicated time in medium containing different carbon sources. (F) Yeast chromatins extracted from WT (LHT001) and H3D/H3H strains were monitored by western blot analysis with antibodies against H3K4ac, H3K4me2, H3K4me3, H3K18ac, H3K27ac, H3K36me3, H3N (H3 N-terminal), H4ac, H4K16ac and H4. Signals are normalized by anti-H4 antibody. (G) Scatter plot showing the average Reads Per Kilobase per Million mapped reads (RPKM) of two replicates distribution of the WT (LHT001) and H3D/H3H strains. The Pearson's product-moment correlation of $Log_{10}$(WT_average_RPKM +1) and $Log_{10}$(H3D/H3H _average_RPKM +1) is 0.9236. The red line is the fitted curve, which has a slope of 0.9966 and which passes through the (0,0) point. R-square is 0.98 and p value≤2.2e-16 (see Materials and methods for details).

DOI: https://doi.org/10.7554/eLife.30178.005

The following source data and figure supplements are available for figure 2:

**Source data 1.** Characterization of the H3D/H3H strain.
DOI: https://doi.org/10.7554/eLife.30178.008
**Source data 2.** Analysis of RNA-Seq data.
DOI: https://doi.org/10.7554/eLife.30178.009
**Figure supplement 1.** Mononucleosome preparation from H3D/H3H cells.
DOI: https://doi.org/10.7554/eLife.30178.006
**Figure supplement 2.** Sample-to-sample reproducibility for the RNA-Seq assay of WT (LHT001) and H3D/H3H strains.
DOI: https://doi.org/10.7554/eLife.30178.007

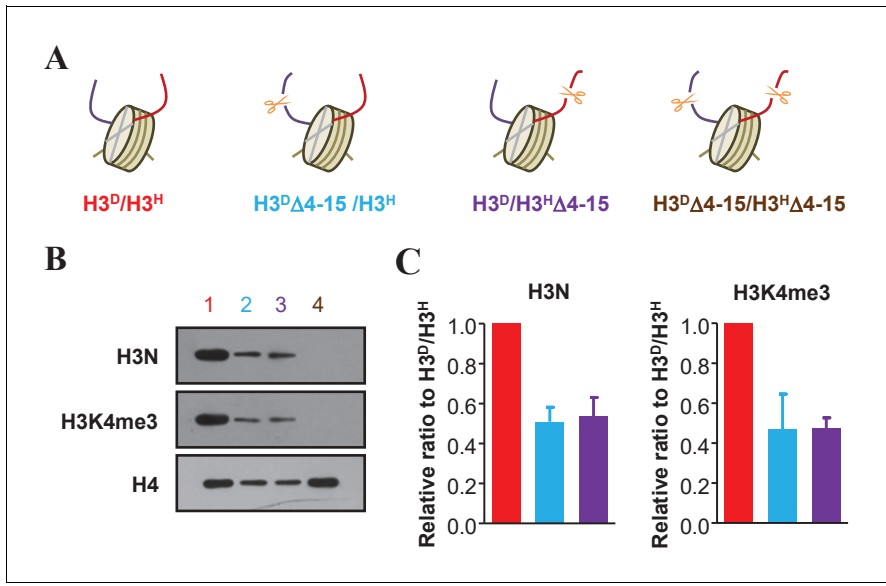

**Figure 3.** Examination of an asymmetric deletion in the N-terminus of histone H3. (A) Schematic illustration of asymmetrical N-terminal-deleted histone H3 mutants of H3DΔ4–15/H3H, H3D/H3HΔ4–15 and H3DΔ4–15/H3HΔ4–15 derived from the H3D/H3H strain. The genotype of each mutant is labeled in a different color, and these colors are applied to identify each mutant in all the panels in this figure. (B) Asymmetric H3 N-terminal-deletion is successfully established on chromatin. Nucleosomes were immunoprecipitated by the anti-H2B antibody from cells of the yeast strains in (A) and analyzed by western blotting using the anti-H3 N-terminal antibody, the anti-H3K4me3 antibody and, as a normalization, the anti-H4 antibody. (C) Quantification of the H3N and H3K4me3 signals in (B) as mean ratio relative to H3D/H3H and normalized to H4 signals. (See the Materials and methods for quantification details.)

DOI: https://doi.org/10.7554/eLife.30178.010

The following source data is available for figure 3:

**Source data 1.** Examination of an asymmetric deletion in the N-terminus of histone H3.
DOI: https://doi.org/10.7554/eLife.30178.011

observed in H3$^D$/H3$^H$ cells (*Figure 3B and C*). These results indicated that H3 N-terminal deletion on one sister H3 did not influence H3K4 methylation on the other.

## K4 me2/3 on sister H3s independently regulates the transcription efficiency of *GAL1* upon induction

As the genes for H3A110D and H3L130H encoded compatible and functional histone H3 proteins, we anticipated that the substitution of K with R on one sister H3 would largely mimic unmethylated K. Thus, asymmetrically modified nucleosomes could be assembled in chromatin in vivo. To test this idea, we first introduced the K4R mutation into H3A110D (H3$^D$K4R) or H3L130H (H3$^H$K4R) in the H3$^D$/H3$^H$ strain (*Figure 4A*). Western blotting showed that H3K4me3 in H3$^D$K4R/H3$^H$ or H3$^D$/H3$^H$K4R cells was approximately 50% lower than that in H3$^D$/H3$^H$ cells, whereas little difference in H3K36me3 was detected among the tested strains (*Figure 4B and C*). Therefore, these results suggest that the hybrid strains contain only mimics of asymmetrically deposited K4me3. For sister H3 histones in a nucleosome, a lack of K4me3 in one tail did not influence K4me3 in the other tail,

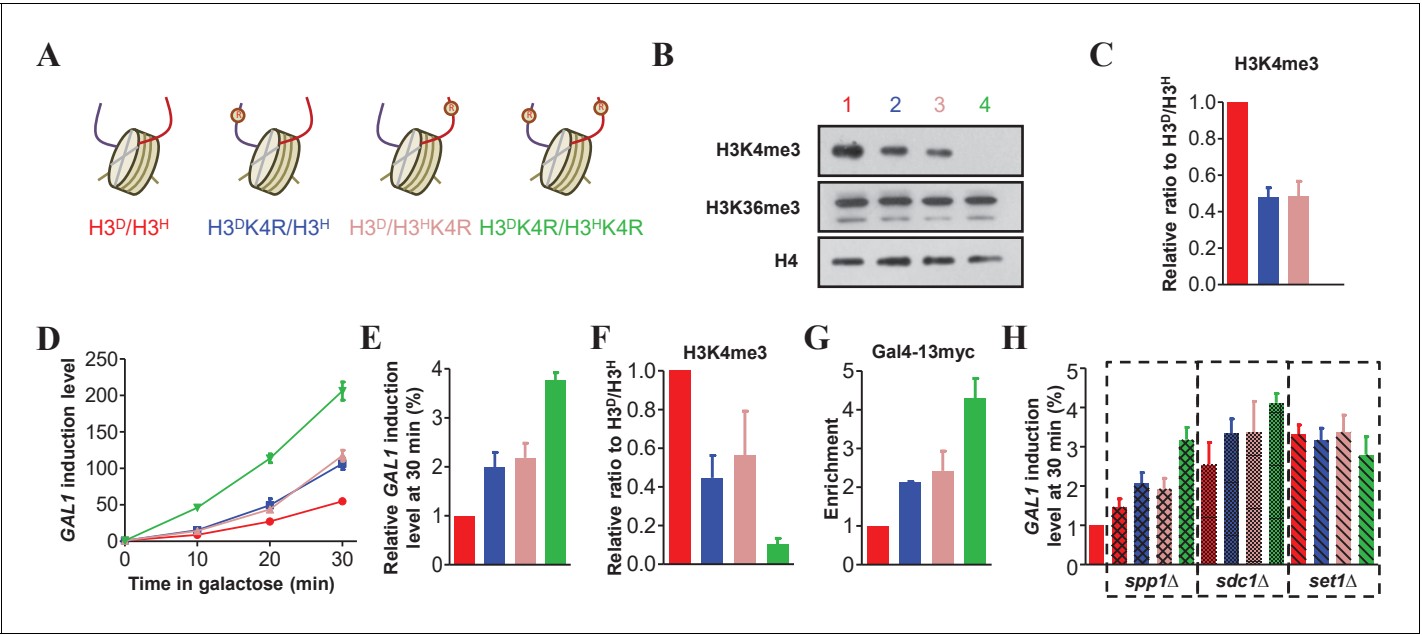

**Figure 4.** Asymmetrically methylated K4 on sister H3s independently upregulates *GAL1* transcription efficiency. (A) Schematic illustration of asymmetrical mutation of histone H3K4R on either H3A110D (H3$^D$) or H3L130H (H3$^H$). The K to R mutation is marked by ® on the histone H3 tails. The color labeling of each strain is applied for all the corresponding panels. (B) Asymmetrical K4me3 marks on sister H3s are successfully established on chromatin. Nucleosomes were immunoprecipitated by the anti-H2B antibody from isogenic strains and analyzed by western blotting using anti-H3K4me3, anti-H3K36me3 and, as a normalization, anti-H4 antibodies. (C) Quantification of western blotting signals for H3K4me3 as mean ratio relative to H3$^D$/H3$^H$ and normalized to signals for H4. (D) The cells with asymmetrical sister K4me3 show an intermediate level of *GAL1* gene expression. Yeast strains illustrated in (A) were subject to galactose induction. Total RNA was prepared at the indicated time points and analyzed by real-time quantitative PCR (qRT-PCR) with primers specific for *GAL1* and normalized by *ACT1*. (E) An alternative representation of the data in (D) at 30 min is expressed as mean ratio to H3$^D$/H3$^H$, whose level was set to 1. (F) Asymmetric K4me3 on sister H3s is established in *GAL1*. ChIP experiments were performed on the promoter of *GAL1* in the indicated yeast strains using the anti-H3K4me3 antibody, and values are normalized to histone H4. (G) The levels of Gal4 recruitment are inversely proportional to the levels of K4me3. Gal4 was tagged with 13 × myc, and ChIP experiments were performed on the UAS region of *GAL1* in the indicated yeast strains with anti-Myc antibody. (H) Detection of *GAL1* levels when *SPP1*, *SDC1* or *SET1* is deleted. *SPP1*, *SDC1* or *SET1*, respectively, was knocked out in the indicated strains, which were subject to galactose induction. RNA was extracted, analyzed and expressed as in (E). H3$^D$/H3$^H$ cells are regarded as WT controls. All of the ChIP values are expressed as mean ratio to H3$^D$/H3$^H$, whose level was set to 1. All error bars indicate s.e.m. for at least duplicated experiments.

DOI: https://doi.org/10.7554/eLife.30178.012

The following source data is available for figure 4:

**Source data 1.** Asymmetrically methylated K4 on sister H3s independently upregulates *GAL1* transcription efficiency.
DOI: https://doi.org/10.7554/eLife.30178.013

consistent with the observation in *Figure 3B*. In addition, H3K4me3 and H3K36me3 were independent of each other because the decrease in H3K4me3 did not alter the level of H3K36me3.

Cells that lack histone H3K4 methylation have an increased *GAL1* induction level (*Pinskaya et al., 2009*). To assess the effect of asymmetrical H3K4me3 on transcription, we assessed *GAL1* mRNA levels in K4R mutant cells. Compared with H3$^D$/H3$^H$ cells, H3$^D$K4R/H3$^H$ and H3$^D$/H3$^H$K4R single-tail mutant cells showed a two-fold increase in *GAL1* mRNA levels. Compared with single-tail mutant cells, H3$^D$K4R/H3$^H$K4R double-tail mutant cells showed a further two-fold increase in *GAL1* mRNA levels (*Figure 4D and E*). *GAL1* mRNA levels were inversely proportional to H3K4me3 levels at the *GAL1* promoter (*Figure 4F*), suggesting a tight correlation between induction levels and H3K4me3 abundance. We also examined the enrichment of Gal4 binding to the *GAL1* promoter using a ChIP assay. Gal4 is the primary activator of *GAL1* transcription (*Johnston, 1987*). A moderate level of Gal4 recruitment to the *GAL1* promoter was observed in the asymmetrical H3$^D$K4R/H3$^H$ and H3$^D$/H3$^H$K4R mutant strains compared with that in their symmetrical H3$^D$/H3$^H$ and H3$^D$K4R/H3$^H$K4R counterparts (*Figure 4G*). Therefore, each K4me3-modified sister histone H3 contributed independently to *GAL1* gene transcription, which is probably recognized and read by the *GAL1* activator Gal4.

Set1C in yeast contains eight subunits, including Set1, Spp1 and Sdc1, and is responsible for methylating histone H3K4 (*Dehé and Géli, 2006*; *Roguev et al., 2001*). Deletion of *SET1* eliminates H3K4 mono-, di- and tri-methylation; deletion of *SPP1* affects only H3K4 tri-methylation; and deletion of *SDC1* affects di- and tri-methylation of H3K4 (*Pinskaya et al., 2009*). To address which type of asymmetrical H3K4 methylation affects *GAL1* transcription, and to confirm that the changes in gene expression were due to asymmetrical H3K4 methylation instead of the K4R mutation, we knocked out *SET1*, *SPP1* and *SDC1* in the H3$^D$/H3$^H$, H3$^D$K4R/H3$^H$, H3$^D$/H3$^H$K4R and H3$^D$K4R/H3$^H$K4R strains and examined *GAL1* levels in galactose medium. As the data show, loss of *SPP1*, *SDC1* and *SET1* led to the upregulation of *GAL1* transcription, which is consistent with previous findings (*Pinskaya et al., 2009*). Meanwhile, an intermediate level of *GAL1* expression was seen in *spp1*Δ H3$^D$K4R/H3$^H$ and *spp1*Δ H3$^D$/H3$^H$K4R cells, whereas no significant difference was found in either *sdc1*Δ or *set1*Δ mutants (*Figure 4H*). As distinguishing between the effects of H3K4me2 and H3K4me3 is difficult, we concluded that H3K4me2/3 but not mono-methylation of H3K4 on sister H3s contributed the most to *GAL1* regulation.

## K36 methylation on sister H3s independently regulates transcription initiation fidelity

As both asymmetrical H3 N-terminal deletion and asymmetrical H3K4me3 were successfully assembled in chromatin, we constructed mutants that mimicked asymmetrical H3K36me. A K36R mutation was introduced into H3A110D (H3$^D$K36R) or H3L130H (H3$^H$K36R) in the H3$^D$/H3$^H$ strain (*Figure 5A*). The level of H3K36me3 and H3K4me3 on chromatin was examined by western blotting. When compared with H3$^D$/H3$^H$ cells, H3$^D$K36R/H3$^H$ or H3$^D$/H3$^H$K36R cells showed an approximately 50% decrease in H3K36me3, whereas little difference in H3K4me3 was detected among the tested strains (*Figure 5B and C*). These data indicated that the H3$^D$K36R/H3$^H$ or H3$^D$/H3$^H$K36R mutants contained only mimics of asymmetrically deposited K36me3, and loss of K36me3 on one tail did not affect K36me3 on the other tail. In addition, in agreement with the data shown in *Figure 4B*, H3K36me3 and H3K4me were independently regulated chromatin modifications.

H3K36me3 directs deacetylation of histone H4 in gene-coding regions to suppress spurious intragenic transcription (*Carrozza et al., 2005*). To address whether H3K36me3 on both sister histone H3s contributed to the regulation of cryptic transcription, we tested the level of intragenic initiation in the H3K36R mutants within the *FLO8*, *PCA1* and *STE11* genes. Each of these genes is regulated by K36 methylation. Northern blot analysis showed that the loss of K36 methylation on H3 tails resulted in short transcripts of the tested genes, consistent with previous findings (*Carrozza et al., 2005*; *Li et al., 2007*). Compared with H3$^D$/H3$^H$ cells and symmetrically mutated H3K36 cells, asymmetrical H3$^D$K36R/H3$^H$ and H3$^D$/H3$^H$K36R cells exhibited an intermediate level of short transcripts (*Figure 5D*). We next used anti-acetylated histone H4 antibodies to perform a ChIP assay on the 3' ORF of the *FLO8*, *PCA1* and *STE11* genes. H4 acetylation (H4ac) levels in H3$^D$K36R/H3$^H$ and H3$^D$/H3$^H$K36R cells were intermediate relative to those in H3$^D$/H3$^H$ cells and symmetrically mutated H3K36 cells. Moreover, H4ac levels were inversely correlated with H3K36me3 levels in the same region (*Figure 5E and F*). In the absence of Set2, the level of H4ac in the tested genes showed no

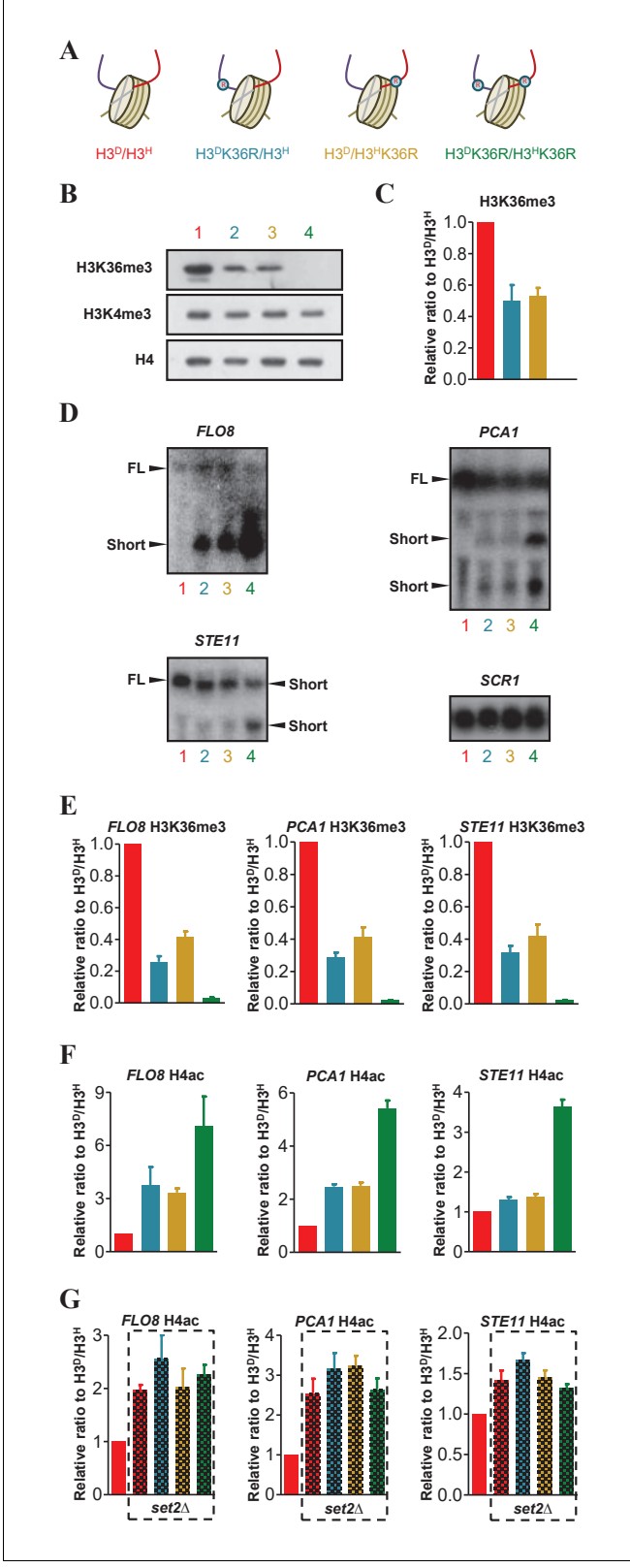

**Figure 5.** The H3K36me3 marks on sister histones regulate transcription independently. (**A**) Schematic illustration of asymmetrical mutation of histone H3K36R on either H3A110D (H3$^D$) or H3L130H (H3$^H$). The K to R mutation is marked by ® on the histone H3. The color labeling of each strain is applied for all the corresponding panels. (**B**) Asymmetrical K36me3 on sister H3s is successfully established on chromatin. Nucleosomes were

*Figure 5 continued on next page*

*Figure 5 continued*

immunoprecipitated by anti-H2B antibody from isogenic strains, as identified by colored numbers, and analyzed by western blot using anti-H3K4me3, anti-H3K36me3 and, as a normalization, anti-H4 antibody. (**C**) Quantification of western blot signals for H3K36me3 as mean ratio relative to H3$^D$/H3$^H$ and normalized to signals for H4. (**D**) Northern blot analysis of the *FLO8*, *STE11* and *PCA1* transcripts in H3K36R mutants. RNA from H3$^D$/H3$^H$, H3$^D$K36R/H3$^H$, H3$^D$/H3$^H$K36R and H3$^D$K36R/H3$^H$K36R strains was probed with sequences complementary to the 3' region of *FLO8*, *STE11*, *PCA1* and, as a loading control, *SCR1*. The full-length (FL) and short transcript signals are indicated. (**E and F**) Asymmetric K36me3 on sister H3s results in an intermediate level of H4ac in *FLO8*, *STE11* and *PCA1*. ChIP experiments were performed in 3' ORF region of *FLO8*, *STE11* and *PCA1* in the indicated cells with anti-H3K36me3 antibody (**E**) and anti-H4ac antibody (**F**). Values are normalized to histone H4 and expressed as mean ratio to H3$^D$/H3$^H$. (**G**) H4ac level of the *FLO8*, *STE11* and *PCA1* loci in *set2Δ* cells bearing different K36me3 states on sister H3s. *SET2* was knocked out in the indicated cells and ChIP experiments were performed as in (**E**). Values are normalized to histone H4 and expressed as mean ratio to H3$^D$/H3$^H$. The H3$^D$/H3$^H$ cells are regarded as a WT control. In all cases, the values of H3$^D$/H3$^H$ are set to 1. All error bars indicate s.e.m. for at least duplicated experiments.

DOI: https://doi.org/10.7554/eLife.30178.014

The following source data is available for figure 5:

**Source data 1.** The H3K36me3 marks on sister histones regulate transcription independently.
DOI: https://doi.org/10.7554/eLife.30178.015

significant differences in H3$^D$/H3$^H$, H3$^D$K36R/H3$^H$, H3$^D$/H3$^H$K36R and H3$^D$K36R/H3$^H$K36R cells (*Figure 5G*). These observations indicated that the regulation of accurate transcription initiation was sensitive to the magnitude of H3K36me3. Accordingly, H4ac levels were regulated by H3K36me3 on both sister histones. In light of these data, we concluded that H3K36me3 on either sister histone played an independent regulatory role in suppressing spurious intragenic transcription.

## K79 methylation on sister H3s cooperatively regulates gene silencing in telomeric regions

H3K79 methylation regulates gene silencing in some telomere-proximal regions (*Takahashi et al., 2011*). To address whether the H3K79 methylation of both sister histones is required to maintain silent chromatin near telomeres, we used strains in which H3K79 could be methylated at either one (H3$^D$K79R/H3$^H$ and H3$^D$/H3$^H$K79R) or none (H3$^D$K79R/H3$^H$K79R) of the H3 sister histones (*Figure 6A*). Western blot analysis revealed that H3K79me2/3 levels in H3$^D$K79R/H3$^H$ and H3$^D$/H3$^H$K79R cells were approximately 50% lower than those in H3$^D$/H3$^H$ cells (*Figure 6B and C*), suggesting the incorporation of asymmetrical H3K79me into chromatin and that the methylation of K79 occurs independently on each sister H3.

We examined the transcription levels of *COS12*, *ERR1* and *ERR3,* which are located proximal to the telomeric ends of chromosomes VIIL, XVR and XIIIR, respectively (*Takahashi et al., 2011*). As expected, the K79R mutations on both sister H3s resulted in decreased silencing of those genes. Surprisingly, H3$^D$K79R/H3$^H$ and H3$^D$/H3$^H$K79R cells containing asymmetrical H3K79me exhibited the same level of silencing loss as that of H3$^D$K79R/H3$^H$K79R or *sir2Δ* cells (*Figure 6D*). A ChIP experiment confirmed that K79me levels at the promoters of the genes tested in H3$^D$K79R/H3$^H$ and H3$^D$/H3$^H$K79R cells decreased to approximately half of those in H3$^D$/H3$^H$ cells (*Figure 6E*). Accordingly, the H4ac level in the ORF region was upregulated in K79R mutated cells (*Figure 6F*). Collectively, these data reveal that K79me marks on both sister H3s act cooperatively to maintain gene silencing near telomeres.

## Cell sensitivity to genotoxic agents is affected by sister histone H3K4, H3K36 and H3K79 modifications

H3K4, H3K36 and H3K79 methylation affects DNA double-strand break (DSB) repair (*Faucher and Wellinger, 2010*; *Jha and Strahl, 2014*; *Pai et al., 2014*). Therefore, we examined the regulatory role of asymmetric H3K4, H3K36 or H3K79 methylation in DSB repair. Mutant cells bearing asymmetrical methylated or non-methylated H3K4, H3K36 or H3K79 were serially diluted and spotted onto plates containing various genotoxic chemicals, including phleomycin, hydroxyurea (HU) or MMS. H3K4R or H3K79R mutations on either one or two sister histones reduced cell growth in the

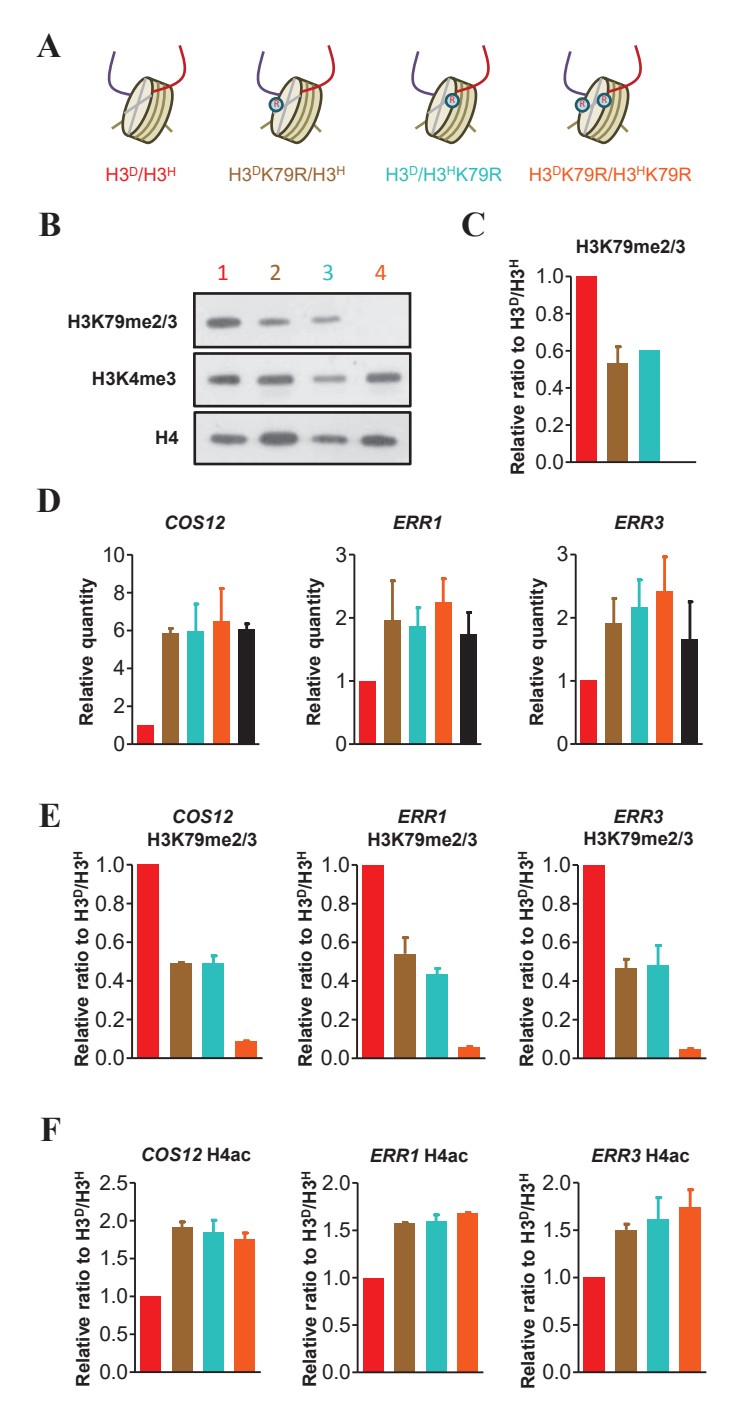

**Figure 6.** The H3K79me marks on both sister histones contribute to the regulation of sub-telomeric chromatin structure. (**A**) Schematic illustration of the asymmetrical mutation of histone H3K79R on either H3A110D (H3$^D$) or H3L130H (H3$^H$). The K to R mutation is marked by ⓇH on the histone H3. The color labeling of each strain is applied in the other panels in this figure. (**B**) Asymmetrical K79me2/3 on sister H3s is established on chromatin. Nucleosomes were immunoprecipitated by anti-H2B antibody from isogenic strains (identified by colored numbers) and analyzed by western blot using anti-H3K79me2/3, anti-H3K4me3 and, as a normalization, anti-H4 antibody. (**C**) Quantification of western blot signals for H3K79me2/3 as mean ratio relative to H3$^D$/H3$^H$. (**D**) Both H3K79me marks on sister H3s are required for the maintenance of telomere silencing. Total RNA was prepared and analyzed by real-time quantitative PCR (qRT-PCR) with primers specific for *COS12*, *ERR1* and *ERR3* and normalized by *ACT1*. The black bar represents data for samples of *sir2Δ* H3$^D$/H3$^H$ mutants, acting as a positive control. Values are

*Figure 6 continued on next page*

*Figure 6 continued*

calculated and expressed as in *Figure 4E*. (**E and F**) Detection of K79me2/3 (**E**) and H4ac (**F**) at the promoters of the *COS12*, *ERR1* and *ERR3*. ChIP experiments were performed on the promoters of *COS12*, *ERR1* and *ERR3* in the indicated cells with anti-H3K79me2/3 antibody (**E**) and anti-H4ac antibody (**F**). Values are normalized to histone H4 and expressed as mean ratio to H3$^D$/H3$^H$. All error bars indicate s.e.m. for at least duplicated experiments.
DOI: https://doi.org/10.7554/eLife.30178.016

The following source data is available for figure 6:

**Source data 1.** The H3K79me marks on both sister histones contribute to the regulation of sub-telomeric chromatin structure.
DOI: https://doi.org/10.7554/eLife.30178.017

presence of the tested genotoxins. Notably, the H3$^D$K36R/H3$^H$K36R mutant was hypersensitive to phleomycin and mildly sensitive to MMS. Compared with the wild type (H3$^D$/H3$^H$) and corresponding double-tail mutant, single-tail H3K36R or H3K79R mutants displayed an intermediate level of sensitivity to the genotoxic agents. The H3K4 mutants showed a similar level of sensitivity to HU and MMS, but single-tail H3K4R mutants displayed less growth in response to phleomycin treatment than did double-tail H3K4R mutants (*Figure 7*).

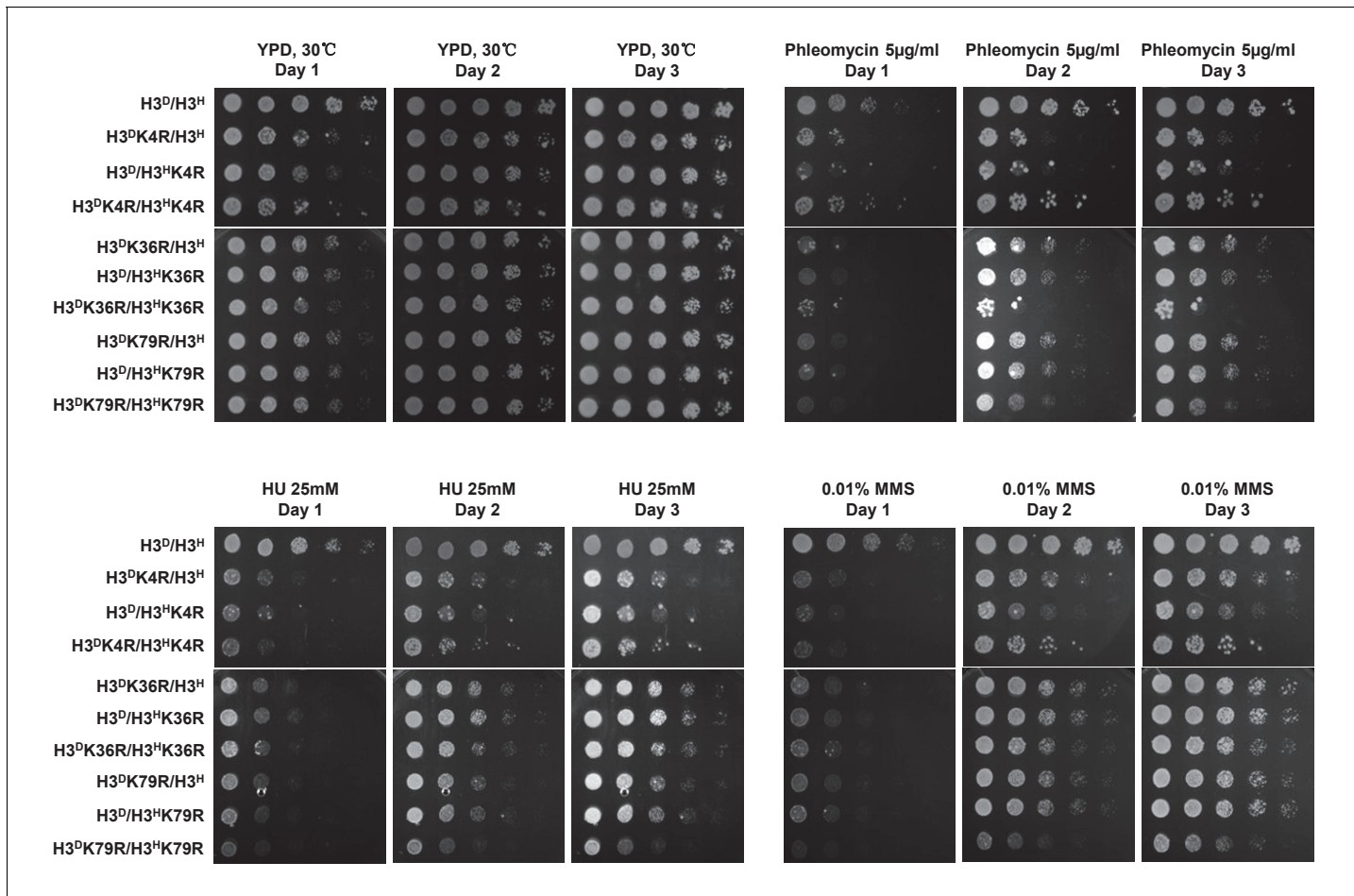

**Figure 7.** The performance of histone mutants challenged by multiple DNA damage reagents. Dotting assay was performed in the indicated mutants as in *Figure 2D*. Plates were photographed after incubation at 30°C on YPD medium or YPD medium containing DNA damage reagents, that is, 5 mg/ml phleomycin, 25 mM HU and 0.01% MMS, for 1, 2 and 3 day(s).
DOI: https://doi.org/10.7554/eLife.30178.018

Together, these observations suggest that, in response to DNA damage, H3K36me and H3K79me marks on sister histones functioned independently, whereas H3K4me marks on sister histones functioned cooperatively. Because the type of DNA damage triggered by the different genotoxic agents and the mechanisms of repair differ, we propose that the combination of sister histone modifications may influence DNA repair in different ways.

## Genome-wide analysis of gene expression in mutants with asymmetrically methylated sister H3K4 under glucose starvation

Chromatin regulators do not appear to affect steady-state transcription, but instead are required for transcriptional reprogramming induced by environmental cues (*Weiner et al., 2012*). For example, the genome-wide gene transcription profile of H3K4A cells was nearly the same as that of WT cells when the cells were grown under normal conditions, whereas differences were observed when the cells were challenged by multiple stress conditions (*Weiner et al., 2012*). To further unravel the genome-wide function of sister H3K4me on transcription, we shifted the cultures of H3K4R mutants and H3$^D$/H3$^H$ strains from 2% to 0.05% glucose in the medium, which mimics calorie restriction. After the cells were grown in 0.05% glucose medium for an hour, we performed RNA-Seq to examine the genome-wide gene-induction profiles, which are presented as fold-change (level of induction). The fold-change value refers to the level of transcription in the induced strains divided by that in the uninduced strains. Of the 6000 genes in the yeast genome, approximately 2500 were altered by the H3K4 to R mutation in response to glucose starvation. Over 1500 genes' fold-change (MID, as defined in Materials and methods) in both asymmetrical K4R mutants (H3$^D$K4R/H3$^H$ and H3$^D$/H3$^H$K4R) fell between those of H3$^D$/H3$^H$ cells and double K4R mutants (*Figure 8—figure supplement 1A*).

Statistical analysis by t-test and a gene skewness score (GSS) model described in the Materials and methods revealed that 22 genes' fold-changes in asymmetrical K4R mutant (H3$^D$K4R/H3$^H$ and H3$^D$/H3$^H$K4R) cells were nearly the same as those in symmetrical K4R mutant (H3$^D$K4R/H3$^H$K4R) cells (*Figure 8A,B*; *Figure 8—figure supplement 1B*, Cluster I), indicating cooperativity of sister K4me at these loci. The fold-changes of 191 genes in asymmetrical K4R mutant (H3$^D$K4R/H3$^H$ and H3$^D$/H3$^H$K4R) cells exhibited an intermediate state between those in symmetrical K4R mutant (H3$^D$K4R/H3$^H$K4R) and WT(H3$^D$/H3$^H$) cells (p<0.05) (*Figure 8A,B*; *Figure 8—figure supplement 1B*, Cluster II), suggesting that K4me on sister H3s independently regulates the expression of these genes. The fold-changes of 158 genes in asymmetrical K4R mutant (H3$^D$K4R/H3$^H$ and H3$^D$/H3$^H$K4R) cells were nearly the same as those in WT H3$^D$/H3$^H$ cells (*Figure 8A,B*; *Figure 8—figure supplement 1B*, Cluster III), indicating redundancy of sister K4me at these loci. An approximately 50% decrease in H3K4me3 in asymmetrical K4R mutants was confirmed in the 5' ORFs of the *YOR008C*, *YMR315W* and *YLR359W* genes, which belong to the three clusters (*Figure 8—figure supplement 1C–E*). These assessments suggest that, in response to glucose starvation stress, H3K4me on two sister histones in different gene loci impose their effects on transcription in a cooperative (e.g., Cluster I), independent (e.g., Cluster II) or redundant (e.g., Cluster III) manner. Interestingly, the genes in Cluster I and II were mostly upregulated (log$_2$fold change >0), while the genes in Cluster III were mostly downregulated (log$_2$fold change <0) (*Figure 8B*), suggesting that under glucose starvation stress, the transcription of upregulated genes may require more subtle regulation mechanisms, such as asymmetrical modification of sister histones.

Although the H3K4R mutation in chromatin largely mimics K4me0, it is not the same as K4me0, and the phenotypes seen in H3$^D$K4R/H3$^H$ and H3$^D$/H3$^H$K4R cells might not result from loss of K4me. To address this issue, we examined the genome-wide gene expression profile of *set1Δ* cells under glucose starvation, and compared its fold-change with that of K4R mutants (*Figure 8—figure supplement 1F*). Many of the genes in Clusters I, II and III overlapped with genes that are regulated by *SET1* deletion (*Figure 8C*, I∩*set1Δ*, II∩*set1Δ* and III∩*set1Δ*, respectively), indicating that these overlapping genes are most probably regulated by K4me rather than the K4R mutation on sister H3s.

## Genes modified by sister H3K4me under glucose starvation are clustered in pathways associated with glycometabolism

To determine which pathways were regulated by asymmetrical K4me on sister H3s in response to glucose starvation, we carried out KEGG pathway analysis (*Huang et al., 2009a*, *2009b*). Nine of

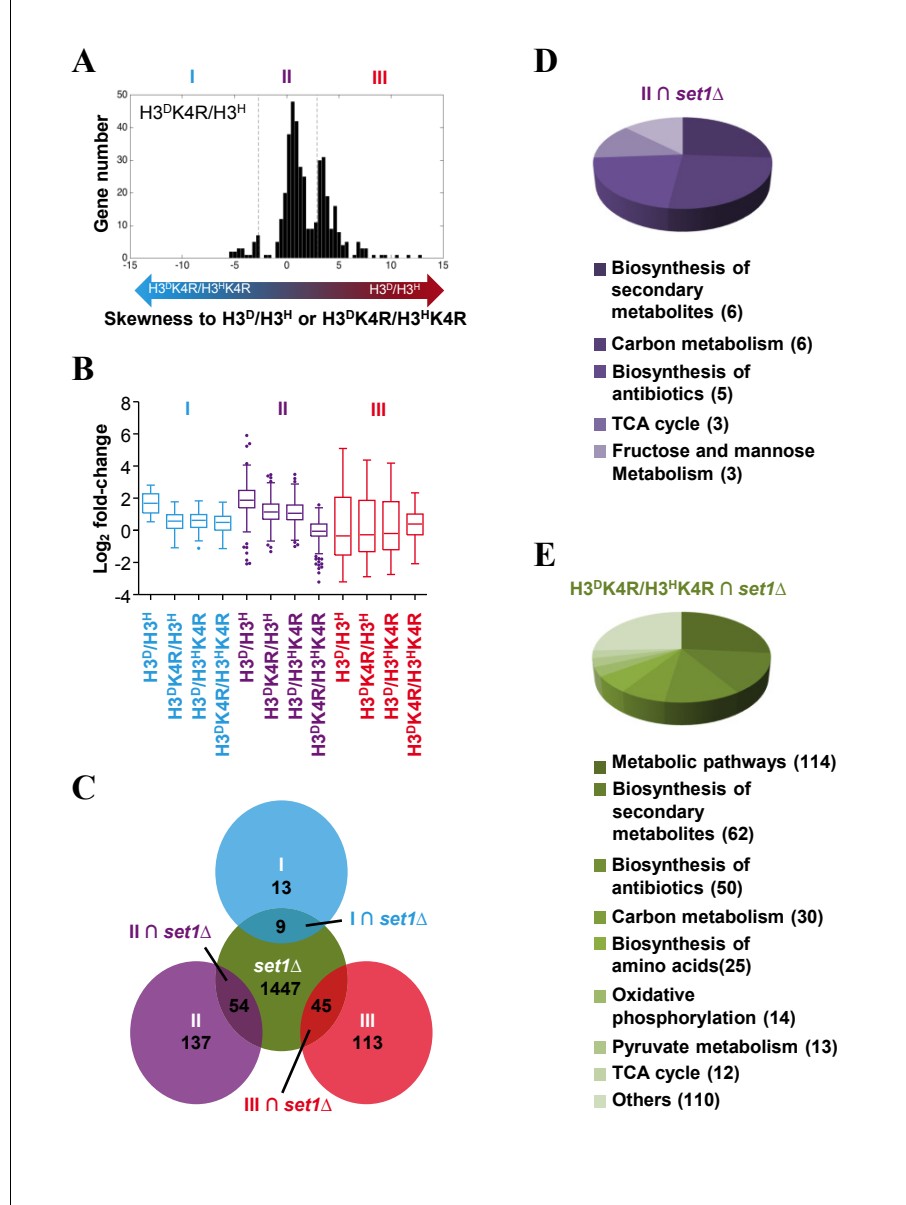

**Figure 8.** Genes in glycometabolism pathways are regulated by asymmetrically methylated K4 on sister H3s in response to glucose starvation. (A) Histogram showing the gene skewness score (GSS) of asymmetrical K4R mutants in set MID calculated by a GSS model (see the Materials and methods for details). Bidirectional arrows with gradient colors indicate the increasing skewness of $\log_2$ fold change in asymmetrical K4R mutants to those in either H3$^D$/H3$^H$ (red end) or H3$^D$K4R/H3$^H$K4R (blue end) cells. Genes are classified to three clusters (I, II and III) by t-test according to the protocols described in the Materials and methods. (B) Boxplots showing the gene's fold-changes of H3$^D$/H3$^H$ and K4R mutants in three gene clusters (blue, cluster I; purple, cluster II; red, cluster III). The $\log_2$ fold-change values are calculated as described in the Materials and methods. (C) Venn diagram showing the overlaps between genes whose fold-change values are significantly altered by *SET1* knockout in the H3$^D$/H3$^H$ strain in response to glucose starvation (green circle), and the genes in clusters I (blue circle), II (purple circle), and III (red circle), respectively. (D and E) Pathways regulated by asymmetrically methylated K4 on sister H3s in response to glucose starvation. Pie charts show the pathways with which the genes are specifically associated, under the regulation of asymmetrical K4me on sister H3s (D), or under the regulation of K4me (E).

DOI: https://doi.org/10.7554/eLife.30178.019

The following source data and figure supplement are available for figure 8:

**Source data 1.** Genes in glycometabolism pathways are regulated by asymmetrically methylated K4 on sister H3s in response to glucose starvation.

*Figure 8 continued on next page*

*Figure 8 continued*

DOI: https://doi.org/10.7554/eLife.30178.021
**Source data 2.** Genes regulated by H3K4R mutation under glucose starvation.
DOI: https://doi.org/10.7554/eLife.30178.022
**Source data 3.** GSS of K4R mutants.
DOI: https://doi.org/10.7554/eLife.30178.023
**Source data 4.** Gene type classification.
DOI: https://doi.org/10.7554/eLife.30178.024
**Source data 5.** KEGG pathway analysis of RNA-Seq data.
DOI: https://doi.org/10.7554/eLife.30178.025
**Figure supplement 1.** Genes in glycometabolism pathways are regulated by asymmetrically methylated K4 on sister H3s in response to glucose starvation.
DOI: https://doi.org/10.7554/eLife.30178.020

the genes in the I⋂*set1Δ* group and 45 of the genes in the III⋂*set1Δ* group could not be mapped to any specific pathways in the KEGG database. Remarkably, 54 of the genes in the II⋂*set1Δ* group, which is regulated by sister H3K4me, were enriched in the pathways involved in glycometabolism, such as carbon metabolism, TCA cycle, and fructose and mannose metabolism (*Figure 8D*). KEGG pathway analysis was performed on genes that were regulated by both the H3$^D$K4R/H3$^H$K4R mutation and *SET1* deletion under glucose starvation. Interestingly, the three pathways involved in glycometabolism in the II⋂*set1Δ* were also found in the list (*Figure 8E*). Therefore, under glucose starvation stress, a significant proportion of H3K4me-responsive genes are regulated by the fluctuation of H3K4me levels on sister H3s. Further analysis of the fold-changes of the genes in the glycometabolism-associated pathways revealed a pattern similar to that of Cluster II (*Figure 8—figure supplement 1G*), suggesting that the independent regulatory mode of sister H3K4me is an important player in response to glucose starvation stress. Collectively, these results support the notions that the on-off regulatory mode for H3K4me is more likely to be applicable to the transcription of genes that do not specifically respond to external stimuli (e.g., genes in the I⋂*set1Δ* and III⋂*set1Δ* groups), whereas the fine-tuning mode evolved to regulate the transcription of genes involved in stress-responsive pathways (e.g., genes in the II⋂*set1Δ* group).

## Discussion

In a nucleosome, two canonical sister histones display identical sequences, suggesting that they have evolved to play similar roles in the regulation of chromatin structure and function. However, similar to post-translational chemical modifications on any protein, the modifications of histones provide an additional layer of chromatin regulation. Paradoxically, sister histones with either asymmetrical modification or coexistence of activation and repression marks have been found in different cell types (*Fisher and Fisher, 2011*; *Mikkelsen et al., 2007*; *Voigt et al., 2012*), raising the possibility that sister histones in a single nucleosome may function independently. In this study, we took advantage of a yeast system that allows for facile genetic manipulation of histones. We identified H3 mutations that prevented homodimer formation and allowed heterodimer formation. After a series of intentional and systematic screenings, we established a bivalent nucleosome system that enabled us to express and monitor sister histone H3s independently in vivo. Owing to the nature of the nucleosome, which is the basic unit structure of chromatin, any possible indirect effect(s) caused by knocking-in a histone mutation cannot be fully excluded from our analysis. Indeed compared with the parental strain, the H3$^D$/H3$^H$ strain did display some minor differences in carbon source preference and nucleosome positioning (e.g., the *GAL1* locus) (*Figure 2C,E*). In addition, when a mutation was introduced into one of the sister histone H3s, only half of the H3 could be modified. Therefore, the global level of modified H3 had a maximum value of 50% of the maximum in a normal cell. Discriminating between the biological consequences caused by forced asymmetrical modification and a forced decrease in total modification is difficult. Nevertheless, we have provided biochemical and functional evidence indicating that this unique genetic system is useful for studying asymmetrical modifications on sister histones.

Using this system, we found that histone H3K4 methylation on one tail is independent of the other tail on the sister H3 histone (*Figure 3B*), suggesting that Set1C binds and modifies one tail in a *cis* fashion. Consistently, mutation of K4R in one of the sister H3s did not affect the methylation of K4 on the other tail (*Figure 4B*). Interestingly, the methylation of K36 or K79 on two sister H3s was also independent (*Figures 5B* and *6B*). The results of our genetic models are consistent with a previous observation that sister histones are not always modified in the same manner simultaneously (*Chen et al., 2011*).

Asymmetrically modified nucleosomes exist on chromatin (*Fisher and Fisher, 2011*; *Mikkelsen et al., 2007*; *Voigt et al., 2012*), but whether these asymmetrical modifications on sister histones function in manner similar to or different from that of symmetrical modifications remains largely unknown. In our study, we observed that K79me on both sister H3 histones was required for silencing telomere-proximal genes through regulation of the acetylation level of histone H4 (*Figure 6D*), establishing a cooperative role for both sister histones in vivo. This mode of regulation was also seen for the genes in the I∩*set1Δ* group when the cells were challenged with glucose starvation (*Figure 8C*). Different from K79me, H3K36me3 on two sister histone H3s did not appear to have a synergistic effect but rather had an additive effect on suppressing spurious transcription (*Figure 5D*), indicating that two K36me3 marks on sister histone H3s altered chromatin structure independently. The same additive effect was observed in the genes grouped in II∩*set1Δ* (*Figure 8C*), as well as in *GAL1* transcription levels (*Figure 4E*). In addition, K4me marks on sister histone H3s redundantly affected the transcription of the genes grouped in III∩*set1Δ* (*Figure 8C*). Consistent with our observations in transcription, sister H3K4me exhibited different regulatory modes in response to various DNA-damage reagents (*Figure 7*). Thus, our data indicate that modifications on sister histones could employ a cooperative, independent, or redundant mode of regulation of chromatin-associated processes. However, why the genes in different loci are subjected to different regulatory mechanisms remains unclear. One possibility is that different gene loci are targeted by different readers, such as activators and repressors that sense the magnitude of H3K4me differently during transcription. This hypothesis is supported by the data in *Figure 4F and G* showing that differential marks of K4me3 on two sister histone H3s affected the enrichment of Gal4-activator binding to the *GAL1* gene promoter, thereby fine-tuning the transcription of *GAL1*. The chromatin readers for different genomic loci have not yet been well characterized, so providing a mechanistic explanation for the different performances of sister histone modifications in every case is difficult.

Histones and their modifications are unique to eukaryotes, and they are important in the packaging of DNA into chromatin. From an evolutionary point of view, it may be that the possesion of two identical copies of each histone in the chromatin in eukaryotes rather than one copy is a sporadic outcome of natural selection. Previous high-throughput analysis showed that epigenetic regulation in the form of histone modification plays a far more pronounced role during gene induction/repression than during steady-state expression (*Weiner et al., 2012*), suggesting the involvement of histone modifications in regulation of gene expression in response to changing environmental cues. In this study, we imposed glucose starvation on yeast cells to mimic an environmental cue. In response, the additive effect of H3K4me on gene transcription was recapitulated in the groups of genes that are enriched in pathways related to glycometabolism, such as carbon metabolism, TCA cycle, and fructose and mannose metabolism (*Figure 8D*). These observations support the notion that in order to adapt to environmental stress, sister histones execute their fine-tuning regulation by differential modifications.

In conclusion, this study provides new insights into how sister histones regulate the plasticity of chromatin structure, as well as gene transcription, and how epigenetic regulation evolves to address variable environmental cues. Given that combinatorial manipulations of sister histone H3 tails have encountered technical challenges in other model systems, the bivalent nucleosome system that we created in this study will be instrumental in further uncovering the role that combinatorial histone H3 modification crosstalk plays in regulating gene expression. In addition, our system for the genetic manipulation of sister histone H3s could be extended to an asymmetry study of sister histone H4s, which have N-terminal tail acetylations representing important epigenetic marks in various biological processes. Moreover, the genetic system that we created will be useful in examining the role that sister histones play in other biological processes, such as DNA repair and recombination, chromatin replication and heterochromatin assembly. Finally, since the protein sequences of histone H3s are highly conserved during evolution, it will be appealing to apply the same scheme to construct a

bivalent nucleosome system in other model systems. However, the challenge might be much greater in higher eukaryotes because the copy numbers of histone genes in these organisms are much higher than those in yeast.

## Materials and methods

### Strains, antibodies and growth conditions

All yeast strains used in this study were derived from yeast strain YPH500 (*Sikorski and Hieter, 1989*). The genotypes of the yeast strains are listed in *Supplementary file 1*. The native promoter of *HHT1* (L130H) in the strains derived from H3$^D$/H3$^H$ was replaced with the *ADE3* promoter. The histone shuffle strain (LHT001) was constructed previously in our lab. Antibodies used in this study are listed in *Supplementary file 2*.

For galactose induction assays, cells were grown in YPD (10 g/L yeast extract, 20 g/L peptone, 2% dextrose) to mid-log phase (OD$_{600}$ = 0.4–0.6) before being shifted to medium containing raffinose (10 g/L yeast extract, 20 g/L peptone, 2% raffinose) overnight. Each sample was induced by 2% galactose for 10–30 min. Remaining samples in raffinose medium were taken as having an induction time of 0 min.

For glucose starvation assays, samples were grown in YPD (2% glucose) to mid-log phase and then shifted to medium containing 0.05% glucose for one hour.

### Mononucleosome preparation and immunoprecipitation

Yeast cells were cross-linked with 1% formaldehyde for 15 min at room temperature and then resuspended in lysis buffer (50 mM HEPES [pH 7.5], 35 mM NaCl, 0.5% Na-Deoxycholate [wt/vol], 5 mM EDTA, 1% Triton X-100, 1 mM phenylmethylsulfonyl fluoride [PMSF], protease inhibitor cocktail). Cells were lysed using glass beads and sonicated to shear the chromatin to fragment sizes of 200–400 bp. After centrifugation at 10,000 g for 10 min, the supernatant fraction was subjected to further fractionation with a 24 ml Superdex-200 column (GE) in IP buffer (10 mM Tris-HCl [pH 8.0], 100 mM NaCl, 0.5 mM EDTA, 1 mM DTT). Fractions containing mononucleosomes were pooled for subsequent incubation with anti-Myc antibody and protein G sepharose beads (GE) overnight at 4°C. The beads were washed with wash buffer (50 mM HEPES [pH 7.5], 150 mM NaCl, 0.5% Na-Deoxycholate [wt/vol], 5 mM EDTA, 1% Triton X-100) and TE (10 mM Tris-HCl [pH 8.0], 1 mM EDTA). Finally, the immunoprecipitated mononucleosomes were eluted from beads with elution buffer (10 mM Tris-HCl [pH 8.0], 1 mM EDTA, 1% SDS [wt/vol]).

### Quantitative reverse transcription-PCR (qRT-PCR)

Total RNA was isolated from yeast cells with an RNeasy mini kit (Qiagen). cDNA was synthesized using the Fastquant RT kit (Tiangen). 1 µl of the RT reaction was used in the subsequent real-time fluorescence quantitative PCR (ABI). Primer pairs used in qRT-PCR were listed in *Supplementary file 3*.

The expression of *GAL1* was normalized to the RNA levels of *ACT1*, and the fold-changes were calculated by defining the relative mRNA level at 0 min as 1.

### MNase digestion assay and Southern blotting

Preparation and digestion of yeast nuclei were performed as described previously (*Kent and Mellor, 1995*; *Wang et al., 2011a*). Yeast genomic DNAs were prepared with phenol-chloroform extraction followed by ethanol precipitation. The DNA was then digested by EcoRI and separated on a 1.6% agarose gel. Digestion patterns were analyzed by indirect-end-labeling. The [$^{32}$P]dCTP incorporated probe whose sequence was listed in *Supplementary file 3* was used for hybridization.

### Preparation of yeast chromatin and chromatin immunoprecipitation (ChIP) assay

Yeast chromatin was prepared as described previously (*Peng and Zhou, 2012*). Specifically, mononucleosomes were purified as described previously for detecting the level of H3N, H3K4me3, H3K36me3 and H3K79me2/3. Chromatin was boiled for 10 min in SDS-PAGE loading buffer and

separated in 15% SDS-PAGE, and then subjected to western blotting. The chromatin immunoprecipitation (ChIP) assay was performed as described previously (*Wang et al., 2011b*).

## Quantification of western blotting

We detected the linear range of all the antibodies. Then we loaded our samples in the linear range and performed a western blot. Quantification of the western blot signals was carried out using ImageJ software (RRID:SCR_003070).

## Northern blot analysis

Total RNA was extracted using the Yeast RNA extraction kit (Qiagen), resolved on agarose-formaldehyde gels and transferred to Hybond-N$^+$ membrane (GE). RNA was crosslinked to the membrane by UV irradiation. Hybridization was carried out in 7% SDS, 1 mM sodium pyrophosphate, 1 M $Na_2HPO_4$, 150 mM $NaH_2PO_4$, and 1 mM EDTA. Probes were generated by PCR.

## RNA-Seq analysis

The method for constructing RNA-Seq libraries was modified from the TruSeq DNA sample preparation kit protocol (Illumina). Briefly, total RNA was isolated using the RNeasy midi kit (Qiagen). The mRNA was purified from total RNA by Dynaloligo(dT) beads (Invitrogen, CA, USA). The first and second strand cDNAs were synthesized using the SuperScript III CellsDirect cDNA Synthesis Kit (Invitrogen) and the SuperScript Double-Stranded cDNA Synthesis Kit (Invitrogen), respectively. The resulting double-stranded DNA was subjected to DNA repair and end-polishing (blunt-end) using the End-It DNA End-Repair Kit (Epicentre). The DNA was then purified with the QIAquick PCR Purification Kit (Qiagen) and a dA-tail was added using the 3'−5' exo-Klenow Fragment (NEB). The resulting purified fragments were ligated to adaptor oligo mix (Illumina) using Quick T4 DNA ligase (NEB). The 200–500 bp ligation products were recovered from a 2% (w/v) agarose gel using the Qiagen gel extraction kit and were PCR amplified with Illumina primers using the KAPA HiFi HotStart kit. The 250–400 bp amplified products were purified again from a 2% agarose gel and used directly for high-throughput sequencing. The raw paired-end reads contained the adapter sequences: the P7 adapter (read1) is 'AGATCGGAAGAGCACACGTCTGAACTCCAGTCAC', the P5 adapter (read2) is 'AGATCGGAAGAGCGTCGTGTAGGGAAAGAGTGT'. We used the FASTX Toolkit (RRID:SCR_005534) to remove the adapter sequences. We trimmed the reads using TopHat (RRID:SCR_013035), only mapping the reads to the transcriptome of sacCer3 (Apr. 2011) with the default parameter. For the mapped reads, we then extracted the reads that have the 'NH:i:1' field. In order to reduce the PCR duplicates' bias, we kept the maximal three records at the same position.

To compare the gene expression profiles between WT (LHT001) and H3$^D$/H3$^H$ strains, the aligned reads were analyzed using Cuffdiff2 (RRID:SCR_001647) (*Trapnell et al., 2012*) to determine the RPKM (Reads Per Kilobase per Million mapped reads) value for each sample. Genes with a change greater than or equal to two folds and p-value $\leq$ 0.001 were regarded as differentially expressed genes and listed in *Figure 2—source data 2*. We identified 406 genes that were downregulated in the H3$^D$/H3$^H$ sample and 243 genes that were upregulated in the H3$^D$/H3$^H$ sample compared with the WT sample. We used FunSpec (RRID:SCR_006952, http://funspec.med.utoronto.ca/) to annotate the differentially expressed genes to get the GO enrichment results (*Robinson et al., 2002*), which were presented in *Figure 2—source data 2*.

For RNA-Seq analysis in glucose starvation experiments, we quantified the number of genes for which at least one read was mapped (RPKM$\neq$0). Fold changes in the transcription of genes under glucose starvation, for genes listed in *Figure 2—source data 2*, were quantified as $FC_{i,j} = log_2((RPKM\_1_{i,j}/RPKM\_1_{act1,j})/(RPKM\_0_{i,j}/RPKM\_0_{act1,j}))$, where $RPKM\_1_{i,j}$ and $RPKM\_0_{i,j}$ refer to RPKM for gene $i$ in sample $j$ after glucose starvation for 1 and 0 hr, respectively. We excluded the gene $i$ when p<0.05 (t-test) by comparing $FC_{i,j}$ in two independent experiments. For the remaining genes, we calculated the average $FC_{i,j}$ (defined as $FCa_{i,j}$) of gene $i$ in sample $j$ using two replicates. To evaluate whether the gene was potentially regulated by H3K4 methylation, we screened gene $i$ of which $FCa_{i,H3^D/H3^H}$ is significantly different (p<0.05) from $FCa_{i,H3^DK4R/H3^HK4R}$ and grouped it to set *DH_4* R4R. Genes in set *DH_4* R4R were listed in *Figure 8—source data 2*.

## Gene skewness score (GSS) model

The skewness of gene transcript fold-change was defined by using the following model. If asymmetrically modified nucleosomes were involved in gene regulation, $FCa_{i,H3^DK4R/H3^H}$ or $FCa_{i,H3^D/H3^HK4R}$ should prerequisitely fall between $FCa_{i,H3^D/H3^H}$ and $FCa_{i,H3^DK4R/H3^HK4R}$. We therefore pooled gene $i$ of set $DH\_4$ R4R into subset $MID$ when $Mid_{i,j} = 1$. The value of $Mid_{i,j}$ was calculated using the following equation:

$$Mid_{i,j} = \frac{\left|FCa_{i,j} - FCa_{i,H3^D/H3^H}\right| + \left|FCa_{i,j} - FCa_{i,H3^DK4R/H3^HK4R}\right|}{\left|FCa_{i,H3^D/H3^H} - FCa_{i,H3^DK4R/H3^HK4R}\right|}$$

The skewness score (GSS) of gene $i$ in sample $j$ was calculated by the equation:

$$GSS_{i,j} = log_2 \frac{\left|FCa_{i,j} - FCa_{i,H3^DK4R/H3^HK4R}\right|}{\left|FCa_{i,j} - FCa_{i,H3^D/H3^H}\right|}, \text{ if } Mid_{i,j} = 1$$

Greater skewness of sample $j$ to H3$^D$/H3$^H$ leads to larger $GSS_{i,j}$. Conversely, greater skewness of sample $j$ to H3$^D$K4R/H3$^H$K4R leads to smaller $GSS_{i,j}$. The results of $Mid_{i,j}$ and $GSS_{i,j}$ were listed in *Figure 8—source data 3*.

## Gene type classification

The gene $i$ in set $MID$ was classified into three subsets: (1) *II*, $FCa_{i,H3^DK4R/H3^H}$ and $FCa_{i,H3^D/H3^HK4R}$ showing significant difference from both $FCa_{i,H3^D/H3^H}$ and $FCa_{i,H3^DK4R/H3^HK4R}$. The maximum and minimum values of $GSS$ in this subset were defined as $GSS_{max}$ and $GSS_{min}$, respectively; (2) *III*, $FCa_{i,H3^DK4R/H3^H}$ and $FCa_{i,H3^D/H3^HK4R}$ exhibiting no difference from $FCa_{i,H3^D/H3^H}$ but significant difference from $FCa_{i,H3^DK4R/H3^HK4R}$ in the condition of each $GSS_{i,H3^DK4R/H3^H}$ and $GSS_{i,H3^D/H3^HK4R}$ larger than $GSS_{max}$; and (3) *I*, $FCa_{i,H3^DK4R/H3^H}$ and $FCa_{i,H3^D/H3^HK4R}$ significantly differing from $FCa_{i,H3^D/H3^H}$ rather than $FCa_{i,H3^DK4R/H3^HK4R}$ in the condition of each $GSS_{i,H3^DK4R/H3^H}$ and $GSS_{i,H3^D/H3^HK4R}$ smaller than $GSS_{min}$. They were listed in *Figure 8—source data 4*.

## Statistics

Data were analyzed by Pearson's product-moment test and Student t-test as indicated.

## Acknowledgements

We thank Drs Brian A Lenzmeier (Buena Vista University) and Hai Jiang for their critical reading of the manuscript. This work was supported by grants from National Natural Science Foundation of China (NSFC 31521061 and 31230040) and Ministry of Science and Technology (MOST 2016YFA0500701 to JQZ). The RNA-Seq data are available from GEO (Gene Expression Omnibus) under accession number GSE104312 (http://www.ncbi.nlm.nih.gov/geo/query/acc.cgi?acc=GSE104312) and GSE88878 (http://www.ncbi.nlm.nih.gov/geo/query/acc.cgi?acc=GSE88878).

## Additional information

### Funding

| Funder | Grant reference number | Author |
| --- | --- | --- |
| National Natural Science Foundation of China | 31521061 | Jin-Qiu Zhou |
| National Natural Science Foundation of China | 31230040 | Jin-Qiu Zhou |
| Ministry of Science and Technology of the People's Republic of China | 2016YFA0500701 | Jin-Qiu Zhou |

The funders had no role in study design, data collection and interpretation, or the decision to submit the work for publication.

## Author contributions
Zhen Zhou, Conceptualization, Resources, Data curation, Software, Formal analysis, Investigation, Methodology, Writing—review and editing; Yu-Ting Liu, Conceptualization, Resources, Data curation, Software, Formal analysis, Investigation, Methodology, Writing—original draft; Li Ma, Resources, Formal analysis, Methodology; Ting Gong, Ling-Li Zhang, Writing—review and editing, Participated in discussion; Ya-Nan Hu, Resources, Methodology; Hong-Tao Li, Chen Cai, Resources, Writing—review and editing; Gang Wei, Resources, Supervision, Methodology; Jin-Qiu Zhou, Supervision, Funding acquisition, Writing—review and editing

## Author ORCIDs
Yu-Ting Liu ⓘ http://orcid.org/0000-0001-7501-7980
Jin-Qiu Zhou ⓘ http://orcid.org/0000-0003-1986-8611

## Decision letter and Author response
Decision letter https://doi.org/10.7554/eLife.30178.034
Author response https://doi.org/10.7554/eLife.30178.035

## Additional files

### Supplementary files
• Supplementary file 1. *S. cerevisiae* strains used in this study.
DOI: https://doi.org/10.7554/eLife.30178.026

• Supplementary file 2. Antibodies used in this study.
DOI: https://doi.org/10.7554/eLife.30178.027

• Supplementary file 3. Oligonucleotides used in this study.
DOI: https://doi.org/10.7554/eLife.30178.028

• Transparent reporting form
DOI: https://doi.org/10.7554/eLife.30178.029

### Major datasets
The following datasets were generated:

| Author(s) | Year | Dataset title | Dataset URL | Database, license, and accessibility information |
|---|---|---|---|---|
| Zhen Zhou, Li Ma, Ya-Nan Hu | 2017 | RNA transcription profile of different yeast mutants under glucose starvation (0.05% glucose) and comparison of transcriptome of WT and H3D_H3H | https://www.ncbi.nlm.nih.gov/geo/query/acc.cgi?acc=GSE104312 | Publicly available at the NCBI Gene Expression Omnibus (accession no. GSE104312) |
| Zhen Zhou, Li Ma, Yu-Ting Liu, Ya-Nan Hu | 2017 | RNA transcription profile of different yeast mutants under glucose starvation (0.05% glucose) | https://www.ncbi.nlm.nih.gov/geo/query/acc.cgi?acc=GSE88878 | Publicly available at the NCBI Gene Expression Omnibus (accession no. GSE88878) |

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
