## [Decision Letter]

Thank you for submitting your article "Modifications on both sister histone H3s coordinate to regulate chromatin structure and transcription" for consideration by *eLife*. Your article has been reviewed by three peer reviewers, one of whom, Timothy Formosa is a member of our Board of Reviewing Editors and the evaluation has been overseen by Jessica Tyler as the Senior Editor.

The reviewers have discussed the reviews with one another and the Reviewing Editor has drafted this decision to help you prepare a revised submission

The authors describe the development and validation of a novel tool for investigating the physiological role of symmetrical modifications of "sister" histone H3 molecules within a given nucleosome in yeast. This system has high potential to probe a set of longstanding questions in this field and the initial results presented here already make a significant contribution. Overall, both the system and the results presented are likely to be of interest to a broad community of chromatin researchers. However, substantial issues need to be resolved prior to publication of the work, and some additional experiments would provide additional support for key conclusions that are currently too preliminary.

Requires further experiments:

1) The authors need to validate more loci; single examples are not sufficient.

The interpretation of the data in Figure 5 is limited due to its examination of a single locus. Similarly, it is premature to conclude from Figure 6 that there are telomeric silencing defects when only a single locus was assessed. The data as presented would also be consistent with promoter-specific effects of the K79 mutations, so analysis of additional loci is required to support the claims put forth. Ideally, a global analysis would be used to obtain the clearest picture, but it should be sufficient to test three independent loci in each case to validate these conclusions.

2) Further genetic analysis of strains would strongly enhance the manuscript.

The authors use screens of plate phenotypes to establish the health of the obligate dimer strain, but one of the potentially most interesting aspects of this study would be to examine the asymmetrical modification strains for these same phenotypes, and this is not included here. It would be very helpful to know how some of the mutations affect phenotypes that reveal changes in gene expression, DNA damage sensitivity, and other features known to be altered by histone modifications. Including this data would increase the impact of the manuscript significantly. Further, as currently presented, the plate phenotypes are difficult to interpret due to the limited growth observed under many conditions even with "WT" cells. Ideally, similar growth rates would be demonstrated by providing growth curves for some crucial conditions, but at the very least the incubation times for the plate assays should be extended, and examination of responses to other challenges such as the Spt- phenotype, caffeine sensitivity, 6AU, rapamycin, synthetic media, raffinose, and galactose should be considered. Given previous results with mutations affecting histone modifications, the Spt- phenotype seems like it would be a very sensitive test of chromatin status, but this requires the use of a strain with a suitable reporter. Screens for some additional phenotypes and better documentation of the plate effected are considered necessary, while examining the effects on reporters like the Spt- phenotype would be strongly encouraged but not required.

3) Additional separation of H3K4 methylation states and more accurate and complete discussion of results.

The experiments in Figure 4 were set up in the text as a dissection of the effects of different methylation states of H3K4, but the results are not analyzed this way. For example, in panel H, if loss of H3K4 methylation leads to activation of the GAL genes, then why does loss of Set1 not show the increase in transcription? Should it not be epistatic with the K4R/K4R strain? There was no explanation for this finding, and the Results section did not adequately address this. Also, Rad6 is an inappropriate mutant to use in this case since it also abolishes H2BK123ub1. There are other mutants that can be used to separate H3K4me states. Further, the fold inductions in panel 4H in the spp1 and rad6 mutants are greater than in the wild-type background (panel 4E), and all induction is lost in the set1 mutant cells. These data are consistent with a model in which H3K4 mono-methylation is required for the induction, but that trimethylation is inhibitory. The authors must compare their observations to previous studies of the role of K4 in gene induction kinetics, and then discuss their results explaining how they contribute to answering the central question posed.

4) As the methylation state of H3K4 is raised as a variable, the authors should either quantify the states separately or explain why this is not necessary.

5) Further exploration of the effects of H3K36 methylation on known downstream pathways is needed.

H3K36 methylation has established roles in suppressing cryptic transcription and in promoting DNA repair, but Figure 5 exclusively examines effects on H4 acetylation as a stand-in for chromatin structure. More complete analysis of potential effects is needed here, and a better description of what is meant by the term "chromatin structure" is required.

6) Further analysis of RNA-seq data; validation of more loci; examination of correlation with transcription level.

Some examples of the results obtained from the RNA-seq experiment in Figure 7 should be validated by qPCR. Representatives of the different categories (at least cluster 1 and cluster 2) should be included. Additionally, analysis of the results to ask whether correlations between the classes defined and the known levels of transcriptional frequency or level of induction should be provided to determine whether these variables explain the classes or their correlations with glycometabolism.

Could be accomplished by text changes or further analysis of existing experiments:

7) Editing for standard English usage is required throughout the manuscript to clarify the claims being made. A professional editing service or assistance from a colleague familiar with the field may be required.

8) In general, the method of normalization for loading and for quantitation should be included in every western blot, with some description of how the authors were able to determine that signals in these blots were in the linear range of detection or were otherwise found to be suitable for quantitation.

9) The MNase sensitivity experiment shows more complexity than is described and the experiment is not clearly explained.

The MNase digestion results presented in Figure 2 are interpreted as evidence that the obligate dimer strain has normal chromatin structure. However, there are substantial differences evident between the wild-type and D-H cells. Most notably, the array of nucleosomes on the GAL1 side of the hypersensitive region produces a much more evenly digested pattern, suggesting altered nucleosome stability, at least in this region, making it an overstatement to claim that there are no structural changes in chromatin resulting from the histone mutations. Further, it is unclear why part of the promoter is labeled as a "hypersensitive region" as this region instead appears refractory to MNase; is the hypersensitivity for DNase I? An additional control using naked DNA would clarify the results shown.

10) The severe growth defect of the delta21-D/delta21-H strain also suggests that the dimerization mutants cause chromatin changes, as deleting the H3 N-termini in otherwise normal strains does not cause this growth defect (e.g. Mann and Grunstein, 1992). The authors should therefore indicate that the D-H mutations cause some changes in chromatin, necessitating that all comparisons must be between D-H cells with wild-type modifiable residues and D-H cells with histone mutations. Along these lines, the authors need to clearly specify which strains are used as controls in each experiment, as "WT" could mean several things, and must be explicit that valid comparisons between appropriately matched controls are being made in every case.

11) The data in Figure 2 are presented as validation that gene expression is not affected by the D-H mutations, but outliers are visible. It would be useful to describe some of them and give statistics for the sample-to-sample reproducibility for a WT strain to give the reader context for understanding how similar the profiles actually are.

12) Figure 3 should be analyzed to provide the ratio of H3K4me3 signals comparing the delta21 and the H-D strains.

13) Figure 4 is lacking H3DK4R/H3HK4R, and 4G is lacking H3D/H3HK4R. These omissions should either be corrected or explained.

14) The H3-N antibody does not recognize the myc-tagged H3, so the authors should explain why it is used instead of some other control for total H3.

15) The results raise an interesting issue regarding the distinction between asymmetrical modification and changes in the total level of modification that should be discussed. For example, in a strain in which only half of the H3 can be modified, the global level of modified H3 now has a maximum value of 50% of the maximum in a normal cell. How can one discriminate between effects due to forced asymmetrical modification and forced decrease in total modification? It is difficult to imagine addressing this experimentally, but the issue should be discussed.

16) Better explanation of the strains used is needed. As noted above, the use of the term "WT" requires explanation in each case. For example, it seems likely that Myc-tagging the H3 protein would have a physiological effect, but no strain with Myc-HHT1 is listed for use as a control for the Myc-tagged heterodimeric nucleosome strains. Is the WT strain the histone shuffle strain? Why is the ADE3 promoter used to drive the Myc-HHT1? Is this a single Myc repeat? It is difficult to determine whether appropriate comparisons are being made without explicit statement of which strains are used in each experiment with a unique name that identifies the strain being described in the strain list provided.

17) Further explanation of the nucleosome purification would be helpful.

Figure 2—figure supplement 1. The DNA fragments look bigger than mononucleosomes; is there other evidence that they are mononucleosomes?

18) The authors refer to the analysis of the δ-21 mutation as "clipping". The term "clipping" is usually used to refer to the natural process of proteolytic tail cleavage, which is induced during sporulation or starvation (Santos-Rosa et al., 2009). In contrast, here the authors are analyzing an engineered gene deletion and are not analyzing cells grown under conditions inducing proteolysis. Therefore, this figure should be titled "Examination of an asymmetric deletion in the N-terminus of histone H3". Furthermore, some of the discussion of this result perpetuates this confusion (subsection “The N-terminal clipping on one of the sister histone H3 tails does not affect the other tail”): "the H3 N-terminal clipping on one of the sister H3s will not influence the clipping on the other one, in other words the endopeptidase activity of cleavage is not affected by losing one of the sister H3 tails." This experiment doesn't include an endopeptidase so this conclusion is not supported by the data.

19) Subsection “K4 methylation on sister H3s independently regulates the transcription efficiency of GAL1 upon induction”, referring to Figure 4: "These results indicate that the nucleosomes with asymmetrical K4me3 is constitutively assembled in chromatin in vivo." The whole cell extracts analyzed show that the steady-state total levels of K4me3-modified H3 is decreased ~50% by the asymmetric mutations, but do not directly measure assembly of those molecules into chromatin in vivo, so the conclusion overstates the observation.

20) In Figure 4, the y-axes of panels D, E and H refer to "rate" but that is not what is being graphed, which is the time course of GAL1 mRNA induction. Rate would be related to the slope of the time course curves. Likewise, the legend for Figure 7—figure supplement 1 does indeed report log2 fold-changes, but not an "induction rate."

---

## [Author Response]

Requires further experiments:1) The authors need to validate more loci; single examples are not sufficient.The interpretation of the data in Figure 5 is limited due to its examination of a single locus. Similarly, it is premature to conclude from Figure 6 that there are telomeric silencing defects when only a single locus was assessed. The data as presented would also be consistent with promoter-specific effects of the K79 mutations, so analysis of additional loci is required to support the claims put forth. Ideally, a global analysis would be used to obtain the clearest picture, but it should be sufficient to test three independent loci in each case to validate these conclusions.

According to the reviewers’ suggestions, we examined three independent loci for K36 mutations and K79 mutations, respectively.

In addition to FLO8, the PCA1 and STE11 genes were tested for intragenic initiation and H4ac levels. We observed an intermediate level of short transcripts in the asymmetrical H3K36R mutants compared with those in H3^D^/H3^H^ cells and symmetrical H3K36R mutants. Consistently, the H4ac levels on the 3' ORF of these genes exhibited similar intermediate phenotypes. Additionally, the intermediate phenotypes of asymmetrical K36R mutants no longer exist when SET2 was knocked out. These data support our conclusion that H3K36me3 on either sister histone makes contributions to suppressing spurious intragenic transcription. These results were presented in Figure 5 in the revised manuscript.

To validate the telomeric silencing defects of the K79R mutations, we detected the transcription level of ERR1 and ERR3 in addition to COS12, which are located proximal to the telomere of chromosome XVR and XIIIR respectively. The H3^D^K79R/H3^H^ and H3^D^/H3^H^K79R cells containing asymmetrical H3K79me exhibited the same level of silencing loss as that of the H3^D^K79R/H3^H^K79R or sir2∆ cells. Accordingly, the H4ac level on the ORF regions of these tested genes were up-regulated in the K79R mutated cells. These analyses support our conclusion that the K79me marks on both sister H3s are indispensible for maintaining silent chromatin near telomeres. These results were presented in Figure 6 in the revised manuscript.

2) Further genetic analysis of strains would strongly enhance the manuscript.The authors use screens of plate phenotypes to establish the health of the obligate dimer strain, but one of the potentially most interesting aspects of this study would be to examine the asymmetrical modification strains for these same phenotypes, and this is not included here. It would be very helpful to know how some of the mutations affect phenotypes that reveal changes in gene expression, DNA damage sensitivity, and other features known to be altered by histone modifications. Including this data would increase the impact of the manuscript significantly. Further, as currently presented, the plate phenotypes are difficult to interpret due to the limited growth observed under many conditions even with "WT" cells. Ideally, similar growth rates would be demonstrated by providing growth curves for some crucial conditions, but at the very least the incubation times for the plate assays should be extended, and examination of responses to other challenges such as the Spt- phenotype, caffeine sensitivity, 6AU, rapamycin, synthetic media, raffinose, and galactose should be considered. Given previous results with mutations affecting histone modifications, the Spt- phenotype seems like it would be a very sensitive test of chromatin status, but this requires the use of a strain with a suitable reporter. Screens for some additional phenotypes and better documentation of the plate effected are considered necessary, while examining the effects on reporters like the Spt- phenotype would be strongly encouraged but not required.

We appreciate this reviewers’ constructive suggestions. We examined the DNA damage sensitivities of H3K4R, K36R and K79R mutants respectively in our system by dotting assay on plate. First of all, consistent with previous findings(Faucher and Wellinger, 2010; Jha and Strahl, 2014; Pai et al., 2014), loss of K4 or K79 modifications on both sister histones exhibited sensitivity to phleomycin, HU and MMS, while losing K36 modifications on both sister histones showed hypersensitivity to phleomycin and mild sensitivity to MMS. These plate phenotypes further confirm that our obligate dimer strain (H3^D^/H3^H^) behaves similar to WT cells (histone shuffle strain, LHT001) in DSB repair. Then we analyzed the plate phenotypes of those asymmetrical modification strains. Intriguingly, we observed that the asymmetrically modified K36 and K79 constantly displayed intermediate levels of sensitivity to all of the genotoxic reagents used, whereas the mutants of H3K4 showed a similar level of sensitivity to HU and MMS, but single-tail H3K4R mutants displayed less growth fitness to the phleomycin treatment compare to the double-tail H3K4R mutant. These observations reveal that in response to DNA damage, the K36me and K79me marks on sister histones functions independently, while the K4me marks on sister histones functions cooperatively. The results were presented in the revised manuscript as Figure 7.

To give a better interpretation of the plate phenotype of WT (histone shuffle strain, LHT001) and H3^D^/H3^H^ cells. We examined their responses to challenges including rapamycin, phleomycin, HU, MMS by dot plate assay (Author response image 1 in this letter; Figure 2 in the revised manuscript) and glycerol, raffinose, galactose by growth curve assay (Figure 2 in the revised manuscript). Meanwhile, the incubation time for the plate assay was extended to 3 days. As expected, H3^D^/H3^H^ and WT strains displayed the same growth phenotype on plate under normal and challenged conditions, as well as the similar growth rates in medium containing glucose and galactose. Whereas, the growth rates of H3^D^/H3^H^ and WT strains differ when they are cultured with glycerol and raffinose as carbon sources. Since we have observed altered nucleosome stability on the GAL1-10 promoter (Figure 2 in the revised manuscript), we suspect that the expression of genes involved in these carbon metabolic pathways are probably disturbed due to altered chromatin structure by our mutations on histone H3. Consistently, when we analyzed the gene expression profile in H3^D^/H3^H^ and WT strains, we observed that the differentially expressed genes are mostly down-regulated in H3^D^/H3^H^ cells. Additionally, the genes encoding cytochrome-c reductase activity and ATPase activity appear to closely related to the growth defect in glycerol medium (Tzagoloff et al., 1975) (Figure 2 and Supplementary file 4 in the revised manuscript).

The Spt- phenotype analysis is a good suggestion, but we don't have the strain with suitable reporters, therefore we didn't include it in our examination. Moreover, caffeine is considered as a “drug”, and it will take months to go through the paperwork to get the reagents. Thus, our experiments did not include caffeine.

**Author response image 1. respfig1:** H3^D^/H3^H^ strain grow as well as WT cell when challenged by rapamycin. WT (LHT001) and H3^D^/H3^H^ cells were grown on YP plate with 100nM rapamycin and photographed at Day1-3 as indicated on the top. Cells grown on YPD plate was set as a control which was displayed in Figure 2 in the revised manuscript.

3) Additional separation of H3K4 methylation states and more accurate and complete discussion of results.The experiments in Figure 4 were set up in the text as a dissection of the effects of different methylation states of H3K4, but the results are not analyzed this way. For example, in panel H, if loss of H3K4 methylation leads to activation of the GAL genes, then why does loss of Set1 not show the increase in transcription? Should it not be epistatic with the K4R/K4R strain? There was no explanation for this finding, and the Results section did not adequately address this. Also, Rad6 is an inappropriate mutant to use in this case since it also abolishes H2BK123ub1. There are other mutants that can be used to separate H3K4me states. Further, the fold inductions in panel 4H in the spp1 and rad6 mutants are greater than in the wild-type background (panel 4E), and all induction is lost in the set1 mutant cells. These data are consistent with a model in which H3K4 mono-methylation is required for the induction, but that trimethylation is inhibitory. The authors must compare their observations to previous studies of the role of K4 in gene induction kinetics, and then discuss their results explaining how they contribute to answering the central question posed.4) As the methylation state of H3K4 is raised as a variable, the authors should either quantify the states separately or explain why this is not necessary.

We apologize that we didn't present our data for Figure 4 clearly enough in our initial submitted manuscript and misled the readers for interpretation of the results. Actually, we knocked out *SET1* in all of the cells presented (H3^D^/H3^H^, H3^D^K4R/H3^H^, H3^D^/H3^H^K4R and H3^D^K4R/H3^H^K4R) and set the value for *set1*∆ H3^D^/H3^H^ to 1. That's also what we did to *SPP1* and *RAD6* knock out groups. In this way, we can only analyze the effect on *GAL1* transcription made by asymmetrical or symmetrical K4R mutations in each knock-out group. And we cannot infer from the data whether loss of these genes bring any effect to *GAL1* transcription because we lack wild-type H3^D^/H3^H^ here as a control.

Aware of this problem, we re-did the GAL1 induction assay in those knock-out mutants together with H3^D^/H3^H^ cells. This time, we set the value for H3^D^/H3^H^ to 1 and the values for all of the knock-out mutants were presented as mean percentage of H3^D^/H3^H^. As the data shown, loss of SPP1, SDC1 and SET1 (thanks to the reviewers’ suggestions, we knocked out SDC1 instead of RAD6 to eliminate K4me2 and me3) all led to up-regulation of GAL1 transcription which is consistent with previous findings (Pinskaya et al., 2009). In the sdc1∆ group, no significant difference was found between sdc1∆ H3^D^/H3^H^ and sdc1∆ H3^D^K4R/H3^H^K4R mutants, indicating that mono-methylation of H3K4 doesn't play a role in regulating GAL1 transcription (to our knowledge, there's no evidence proving that H3K4 mono-methylation is required for GAL1 induction previously). Meanwhile, the intermediate level of GAL1 expression was seen in the spp1∆ H3^D^K4R/H3^H^ and spp1∆ H3^D^/H3^H^K4R cells while no significant difference was found in the set1∆ mutants. Since it is difficult to distinguish between the effects of H3K4me2 and H3K4me3, we conclude that H3K4me2/3 on sister H3s may contribute to GAL1 regulation in an independent mode. The results were presented in the revised manuscript as Figure 4.

The reviewers are quite right that it's better to quantify the states of K4 methylation separately when we try to separate the function of them. However, when we detected the linear range of our anti-H3K4me2 and anti-H3K4me1 antibodies, we found that the distributions of the signals are not linear and the signal is not in proportional to the quantity of loading (Author response image 2). Therefore., these antibodies cannot be used for quantification assay. By the way, since we excluded the contribution of K4me1 in GAL1 induction and cannot separate the function of K4me2 and me3, lacking K4me1 and me2 quantification will not interfere our conclusion. But, yes, it would be better if we can find an appropriate anti-H3K4me2 antibody for quantification.

5) Further exploration of the effects of H3K36 methylation on known downstream pathways is needed.H3K36 methylation has established roles in suppressing cryptic transcription and in promoting DNA repair, but Figure 5 exclusively examines effects on H4 acetylation as a stand-in for chromatin structure. More complete analysis of potential effects is needed here, and a better description of what is meant by the term "chromatin structure" is required.

Thanks to the reviewers for their critical comments. Although hyper H4 acetylation level suggest a loose chromatin, it seems too preliminary to regard H4 acetylation phenotypes as a stand-in for chromatin structure. It's more reasonable to interpret it as a downstream effect by K36 methylation loss in regulation of cryptic transcription as previously reported. Following this suggestion, we performed northern blot assay to examine the level of intragenic initiation in the H3K36R mutants within the *FLO8, PCA1* and *STE11* genes which have been demonstrated to be regulated by K36 methylation (Carrozza et al., 2005; Li et al., 2007). An intermediate level of short transcripts was observed in asymmetrical H3K36R mutants compared with those in H3^D^/H3^H^ and symmetrically mutated H3K36 cells, which is consistent with what we have observed initially for H4ac. These results were presented in Figure 5 in the revised manuscript. Accordingly, we have revised the statement in the revised manuscript: sister H3K36me3 plays an independent regulatory role in suppressing spurious intragenic transcription and the underlying mechanism probably relies on the regulation of H4ac abundance.

According to the reviewers’ suggestions, we examined the effects of asymmetrical K36me in DSB repair. We tested the sensitivities to various genotoxic reagents including phleomycin, HU and MMS in mutants bearing asymmetrical methylated K36, and compared the phenotypes with its symmetrical methylated and non-methylated counterparts. Consistent with previous findings (Jha and Strahl, 2014; Pai et al., 2014), loss of K36 modifications on both sister histones showed hypersensitivity to phleomycin and mild sensitivity to MMS. The cells containing asymmetrically modified K36 displayed an intermediate level of sensitivity to phleomycin and MMS, suggesting that K36 modifications on sister histone H3 may regulate DSB repair in an independent mode. These results were presented in Figure 7 in the revised manuscript.

6) Further analysis of RNA-seq data; validation of more loci; examination of correlation with transcription level.Some examples of the results obtained from the RNA-seq experiment in Figure 7 should be validated by qPCR. Representatives of the different categories (at least cluster 1 and cluster 2) should be included. Additionally, analysis of the results to ask whether correlations between the classes defined and the known levels of transcriptional frequency or level of induction should be provided to determine whether these variables explain the classes or their correlations with glycometabolism.

According to the reviewers’ suggestions, we validated the RNA-seq experiment by performing qPCR in the 5' ORFs of YOR008C, YMR315W and YLR359W genes which belong to the Cluster I, II, III respectively. And these results were presented in Figure 8—figure supplement 1 in the revised manuscript.

The known level of induction is the prerequisite for defining our three clusters. By comparing the level of induction (expressed as log_2_fold-change in our manuscript) in the asymmetrical K4R mutants with that in the H3^D^/H3^H^ cells and in the symmetrical K4R mutants, we defined three clusters as: Cluster I, the level of induction in the asymmetrical K4R mutants is the same with that in the symmetrical K4R mutants, but different from that in the H3^D^/H3^H^ cells; Cluster II, the level of induction in the asymmetrical K4R mutants is intermediate between that in the H3^D^/H3^H^ cells and in the symmetrical K4R mutants; Cluster III, the level of induction in the asymmetrical K4R mutants is the same with that in the H3^D^/H3^H^ cells, but different from that in the symmetrical K4R mutants. These correlations between clusters and level of induction help us to infer that the transcription of genes in Cluster I is hypersensitive to the loss of one H3K4me on sister H3s, and one K4me on sister H3s is insufficient to exert any effect; the transcription of genes in Cluster II is sensitive to the magnitude of H3K4me, and is regulated by sister histones through H3K4 methylation; the transcription of genes in Cluster III is hyposensitive to the loss of one H3K4me on sister H3s, and the function of K4me on sister H3 is redundant. Further KEGG pathway analysis showed that genes regulated by H3K4me are involved in pathways directly responsive to glucose starvation, e.g. carbon metabolism, the TCA cycle, and fructose and mannose metabolism (Figure 8 in the revised manuscript). Analysis of the fold-change of those genes which are involved in the three pathways revealed a similar pattern to that of Cluster II (Figure 8—figure supplement 1 in the revised manuscript). Therefore, under glucose starvation stress a significant proportion of glycometablism associated genes are regulated by the fluctuation of H3K4me level on sister H3s. According to the reviewers’ suggestions, we have revised some statements of the RNA-seq part in our manuscript.

Could be accomplished by text changes or further analysis of existing experiments:7) Editing for standard English usage is required throughout the manuscript to clarify the claims being made. A professional editing service or assistance from a colleague familiar with the field may be required.

According to this reviewers’ suggestion, we asked Dr. Brian Lenzmeier, who is a former colleague of mine, and now a professor of biology in Buena Vista University (Storm Lake, IA 50588, USA), to help edit the manuscript.

8) In general, the method of normalization for loading and for quantitation should be included in every western blot, with some description of how the authors were able to determine that signals in these blots were in the linear range of detection or were otherwise found to be suitable for quantitation.

We have included the method of normalization for loading and quantitation in the figure legends and Methods respectively.

To determine that signals in these blots were in the linear range of detection and suitable for quantitation, we provided the linear range of the antibodies used for quantitation in this letter as Author response image 2 and we loaded our samples in the linear range.

9) The MNase sensitivity experiment shows more complexity than is described and the experiment is not clearly explained. The MNase digestion results presented in Figure 2 are interpreted as evidence that the obligate dimer strain has normal chromatin structure. However, there are substantial differences evident between the wild-type and D-H cells. Most notably, the array of nucleosomes on the GAL1 side of the hypersensitive region produces a much more evenly digested pattern, suggesting altered nucleosome stability, at least in this region, making it an overstatement to claim that there are no structural changes in chromatin resulting from the histone mutations. Further, it is unclear why part of the promoter is labeled as a "hypersensitive region" as this region instead appears refractory to MNase; is the hypersensitivity for DNase I? An additional control using naked DNA would clarify the results shown.

Thanks to the reviewers for their great insight. Since the signals of our initial MNase digestion results were a little weak in the detected regions, we repeated the MNase digestion assay (presented as Figure 2 in our revised manuscript). As this reviewer has pointed out, indeed we observed substantial differences between the wild-type and D-H cells on the GAL1 side. The nucleosome array was much more unstable in D-H cells in this region. Therefore, we changed our statement in our revised manuscript for this part.

This reviewers are right that the "hypersensitive region" is defined relative to chromatin digestion by DNase I (Lohr, 1993), thus was labeled accordingly either in DNase I digestion result or in MNase digestion result (Lohr and Lopez, 1995). To avoid confusion, we relabeled this region as "UAS" which is reported to be included in the hypersensitive region and protected by effecter proteins (such as GAL4) from digesting (Lohr and Lopez, 1995).10) The severe growth defect of the delta21-D/delta21-H strain also suggests that the dimerization mutants cause chromatin changes, as deleting the H3 N-termini in otherwise normal strains does not cause this growth defect (e.g. Mann and Grunstein, 1992). The authors should therefore indicate that the D-H mutations cause some changes in chromatin, necessitating that all comparisons must be between D-H cells with wild-type modifiable residues and D-H cells with histone mutations. Along these lines, the authors need to clearly specify which strains are used as controls in each experiment, as "WT" could mean several things, and must be explicit that valid comparisons between appropriately matched controls are being made in every case.

Thanks to the reviewers for their critical comments. In the work by Mann and Grunstein, they performed various kinds of N-terminal deletions of histone H3 while the first three amino acids were left to promise normal post-translational processing of the N-terminal residue and stability (Mann and Grunstein, 1992). As cited in their paper, it has been reported that the amino-terminal residues of a protein are important for its stability (Bachmair et al., 1986). We apologize that we didn't consider this issue in our initial design for the N-terminal deletion strategy. That's why we observed severe growth defect of the delta21-D/delta21-H strain. Therefore, we changed our N-terminal deletion strategy to deletion of the N-terminal 15 amino acids while keeping the first three amino acids (Δ4-15). It turned out that the growth defect is remarkably alleviated, while a little slow growth was still observed in the N-terminal deletion mutants which is consistent with previous findings that the doubling time in Δ4-15 strain is prolonged (Mann and Grunstein, 1992). Accordingly, we changed the figure and content for this part from Δ21 to Δ4-15 in our revised manuscript (refer to Figure 3 in our revised manuscript).

We used histone shuffle strain LHT001 as WT control in the characterization of the D-H cells. In the investigation of the function of asymmetrical modifications, the D-H cell was regarded as WT control. We have specified these in our revised figure legends and manuscript.

**Author response image 2. respfig2:** Linear range of antibodies. Nucleosomes were immunoprecipitated and loaded in gradient as indicated. Western blot was performed using the indicated antibodies and quantified by ImageJ software. The relationship (black line) between sample loading and fluorescence signal was displayed by linear regression. The theoretical correlation between these two variables was represented as red dash line.

11) The data in Figure 2 are presented as validation that gene expression is not affected by the D-H mutations, but outliers are visible. It would be useful to describe some of them and give statistics for the sample-to-sample reproducibility for a WT strain to give the reader context for understanding how similar the profiles actually are.

The description of some of the outliers was added in our revised manuscript. The statistical analysis for the sample-to-sample reproducibility for WT and H3^D^/H3^H^ strains were presented in the revised Figure 2—figure supplement 2.

12) Figure 3 should be analyzed to provide the ratio of H3K4me3 signals comparing the delta21 and the H-D strains.

The ratio of H3K4me3 signals comparing the N-terminal deletion mutants and the H-D strains was provided in our revised Figure 3.

13) Figure 4 is lacking H3DK4R/H3HK4R, and 4G is lacking H3D/H3HK4R. These omissions should either be corrected or explained.

These omissions were corrected in our revised Figure 4.

14) The H3-N antibody does not recognize the myc-tagged H3, so the authors should explain why it is used instead of some other control for total H3.

Because both the H3^D^ and H3^H^ mutations are at the C-terminus of histone H3, the anti-H3-C antibody cannot recognize our mutated histone H3^D^ and H3^H^. We chose the anti-H3-N antibody for detection. We didn't set total H3 control here, because we normalized the signals of the immunoprecipitated Myc- H3^H^ and Myc- H3^H^ by myc antibody (compare the Myc panel of lane 4 and 6) and detected their co-immunoprecipitated counterparts (untagged H3^D^ and untagged H3^H,^respectively) by H3-N antibody (compare the H3N panel of lane 4 and 6).

15) The results raise an interesting issue regarding the distinction between asymmetrical modification and changes in the total level of modification that should be discussed. For example, in a strain in which only half of the H3 can be modified, the global level of modified H3 now has a maximum value of 50% of the maximum in a normal cell. How can one discriminate between effects due to forced asymmetrical modification and forced decrease in total modification? It is difficult to imagine addressing this experimentally, but the issue should be discussed.

In a normal cell, down-regulation of the total level of modification definitely exists as a mode to regulate chromatin associated processes. However, it's difficult to achieve a fixed value (such as 50%) for a certain modification in a normal cell experimentally. Therefore, it's rather difficult to investigation the relationship between the magnitude of histone modification and its effects. In our system, we forced the establishment of asymmetrical modifications on chromatin to have a maximum value of 50% of the maximum for the global level of modified H3. We found that some cellular processes are sensitive to the decrease of modification level (the effects of 50% modification loss equals that of 100% loss), some are not sensitive (the effects of 50% modification loss equals that of 0% loss), while some are fine-tuned by 50% modification loss. Since multiple asymmetrical modifications have been found in cells, they should contribute to regulate cellular processes. Based on previous observations by many labs, it's speculated that genes possessing asymmetrical modifications are in a poised state, enabling them to be rapidly activated upon suitable developmental cues and/or environmental stimuli (Voigt et al., 2013). Consistently, in our story, we found that most of the genes fine-tuned by asymmetrical modifications are closely related to stress-response. Thus, we conjecture that oscillation of the total level of histone modification may apply to situations requiring rough regulation (on-off regulation) while the role of asymmetrical modification may be related to situations calling for subtle regulation (fine-tuning regulation). It'll be perfect if we can come up with a strategy to control the level of forced decrease in total modification. We discussed this issue in the "Discussion" part of our revised manuscript.

16) Better explanation of the strains used is needed.As noted above, the use of the term "WT" requires explanation in each case. For example, it seems likely that Myc-tagging the H3 protein would have a physiological effect, but no strain with Myc-HHT1 is listed for use as a control for the Myc-tagged heterodimeric nucleosome strains. Is the WT strain the histone shuffle strain? Why is the ADE3 promoter used to drive the Myc-HHT1? Is this a single Myc repeat? It is difficult to determine whether appropriate comparisons are being made without explicit statement of which strains are used in each experiment with a unique name that identifies the strain being described in the strain list provided.

Thanks to the reviewers for their critical comments. As suggested, we have included explanation for the use of "WT" and marked it by the unique name described in the strain list in each case in our revised manuscript and figure legends. We feel sorry that we omitted the information for strain with Myc-HHT1 in our initial strain list, and it's added in our revised list. In the part of "Characterization of the H3^D^/H3^H^ strain", we use the term of "WT" and it's indeed the histone shuffle (Agez et al., 2007)strain (LHT001 in the strain list). All of the histones tagged with Myc in our story refer to tagging with a single Myc repeat, which is illustrated in the revised strain list.

The main purpose of using the ADE3 promoter to drive one of the copy of HHT1 gene is to reduce the expression of H3^H^ (including myc-H3^H^) expression. The reason is as following. When we examined the incorporation ratio of H3^D^ and H3^H^ in chromatin, we constructed two strains: in one strain, the H3^D^ was tagged with Myc-HA and the H3^H^ was tagged with Myc; in the other strain, the H3^D^ was tagged with Myc and the H3^H^ was tagged with Myc-HA. The Myc and Myc-HA tags are used to discriminate the size of H3^D^ and H3^H^ proteins. In the immunoprecipation experiment, we immunoprecipitated the mono-nucleosomes using anti-H2B antibody, and examined Myc- and Myc-HA-tagged H3^D^ and H3^H^ by anti-Myc antibody. We observed severe decrease of expression levels for Myc-HA-tagged proteins (Attached Figure 4, the first two lanes). Under these circumstances, the IPed signal for Myc-HA-H3^H^ is far much higher than that for Myc-H3^D^, suggesting that undesired H3^H^- H3^H^ dimer was incorporated into nucleosome. In contrast, when the expression level of Myc-HA-H3^D^ is much higher than Myc-H3^H^ (Attached Figure 4, the last lane), the incorporation ratio of H3^D^ and H3^H^ was near 1:1. We suspect that though mutant strains containing only H3^H^ cannot survive on the FOA plate (Figure 1 in our revised manuscript), there could be very weak interactions between two H3^H^ moleculeswhen the expression of H3^H^ is too high due to the strong natural promoter of H3^H^. This specaulation is reasonable. Because (1) the pKa value for the side chain of histidine (H) is 6.02 (according to the evaluation " pH = pKa + lgBHB+ "), under physiological pH conditions which is around 6, there could be 1molecule of uncharged histidine out of 2 molecules of histidine. Accordingly, the pKa value for the side chain of Aspatate (D) is 3.71 (according to the evaluation " pH = pKa + lgA-HA "), there could be 1molecule of uncharged aspatate out of 200 molecules of aspatate. Therefore, the probability of interaction between H3^H^ molecules should be much higher than that between H3^D^ molecules. (2) histone chaperone Asf1 has been found to prefer binding hydrophobic groups (Agez et al., 2007; Antczak et al., 2006). The uncharged histidine exhibits relatively strong hydrophobicity, thus is more preferred by Asf1.

In order to alleviate the risk of H3^H^- H3^H^ dimer formation, we used ADE3 promoter, which is much weaker than the natural promoterof HHT1, to drive H3^H^ expression (Ghaemmaghami et al., 2003; Kulak et al., 2014), while H3^D^ was still driven by its natural promoter. In this reformed strain, we performed IP and ChIP assays and showed that H3^D^ and H3^H^ are assembled into nucleosomes at a ratio of 1:1 in total level and at specific gene loci (e.g. GAL1-10 promoter) respectively (Figure 2 in our revised manuscript). We indicated the change of promoter in the H3^D^/H3^H^ strain and briefly explained the reason in our revised manuscript.

**Author response image 3. respfig3:** Immunoprecipitation analysis using anti-Myc antibody.

17) Further explanation of the nucleosome purification would be helpful.Figure 2—figure supplement 1. The DNA fragments look bigger than mononucleosomes; is there other evidence that they are mononucleosomes?

In the preparation of mononucleosome, we used sonication to break the chromatin. In this way, we got mononulcosomes with flexible linker DNA ends which may leads to bigger DNA fragments than that in theory. By the way, we compared our FPLC spectra for purification of mononucleosomes with the reference spectra of our chromatographic column provided by GE healthcare. And the fractions we collected were peaked within the theoretical size range for mononucleosome (Figure 2—figure supplement 1 in our revised manuscript).

18) The authors refer to the analysis of the δ-21 mutation as "clipping". The term "clipping" is usually used to refer to the natural process of proteolytic tail cleavage, which is induced during sporulation or starvation (Santos-Rosa et al., 2009). In contrast, here the authors are analyzing an engineered gene deletion and are not analyzing cells grown under conditions inducing proteolysis. Therefore, this figure should be titled "Examination of an asymmetric deletion in the N-terminus of histone H3". Furthermore, some of the discussion of this result perpetuates this confusion (subsection “The N-terminal clipping on one of the sister histone H3 tails does not affect the other tail”): "the H3 N-terminal clipping on one of the sister H3s will not influence the clipping on the other one, in other words the endopeptidase activity of cleavage is not affected by losing one of the sister H3 tails." This experiment doesn't include an endopeptidase so this conclusion is not supported by the data.

According to the reviewers’ suggestions, we changed the statements from "clipping" to "N-terminal deletion" in our revised manuscript and figure legends.

19) Subsection “K4 methylation on sister H3s independently regulates the transcription efficiency of GAL1 upon induction”, referring to Figure 4: "These results indicate that the nucleosomes with asymmetrical K4me3 is constitutively assembled in chromatin in vivo." The whole cell extracts analyzed show that the steady-state total levels of K4me3-modified H3 is decreased ~50% by the asymmetric mutations, but do not directly measure assembly of those molecules into chromatin in vivo, so the conclusion overstates the observation.

According to the reviewers’ suggestions, we changed the statements to "These results suggest asymmetrical K4me3 has been mimicked on chromatin in vivo.”

20) In Figure 4, the y-axes of panels D, E and H refer to "rate" but that is not what is being graphed, which is the time course of GAL1 mRNA induction. Rate would be related to the slope of the time course curves. Likewise, the legend for Figure 7—figure supplement 1 does indeed report log2 fold-changes, but not an "induction rate."

Thanks to the reviewers’ suggestions, we changed the label of y-axes in Figure 4 to "*GAL1* induction level" and deleted the "induction rate" in the legend for Figure 8 (initial Figure 7) in our revised manuscript.